# Phase separation of BuGZ regulates gut regeneration and aging through interaction with m⁶A regulators

Qiaoqiao Zhang[1,2,3], Kai Deng[4,5], Mengyou Liu[1], Shengye Yang[1], Wei Xu[1], Tong Feng[3], Minwen Jie[1], Zhiming Liu[2], Xiao Sheng[2], Haiyang Chen[2] ✉ & Hao Jiang[1] ✉

Exploring the role of phase separation in intracellular compartment formation is an active area of research. However, the associations of phase separation with intestinal stem cell (ISC)-dependent regeneration and aging remain unclear. Here, we demonstrate that BuGZ, a coacervating mitotic effector, shows age- and injury-associated condensation in *Drosophila* ISC nuclei during interphase. BuGZ condensation promotes ISC proliferation, affecting *Drosophila* gut repair and longevity. Moreover, m⁶A reader YT521-B acts as the transcriptional and functional downstream of BuGZ. The binding of YT521-B promotor or m⁶A writer Ime4/ Mettl14 to BuGZ controls its coacervation, indicating that the promotor may accelerate the phase transition of its binding transcription factor. Hence, we propose that phase separation and m⁶A regulators may be critical for ameliorating ISC-dependent gut regeneration and aging and requires further study.

Biological liquid-liquid phase separation (LLPS) triggers the formation of condensed liquid-like droplets, resembling intracellular compartmentations that spatiotemporally facilitate or attenuate biochemical reactions[1,2]. The biophysical properties of these macromolecular condensates are precisely controlled by their concentrations, structural features including intrinsically disordered regions (IDRs) and post-translational modifications (PTMs), contiguous environment and interacting chaperones containing DNA, RNA, and proteins[3–5]. DNA-protein complexes such as centrosomes, mitotic spindle, actin polymerization complex, telomere, chromatin, heterochromatin, and nucleolus form droplet-like condensates[6,7]. Despite super-enhancer, enhancer, exogenous pathogen, and telomere DNA have been shown to induce phase separation of their corresponding associated

proteins[7,8], whether the promotor influences the phase transition of its targeted transcription factor (TF) during the transcription process remains ambiguous. Biological functions and impacts of phase separation have been ill-defined and fueled by the ease of visualizing the droplet-shaped coacervations and disrupting the reversible condensate assembly in vivo and in vitro, which have advanced various biological fields[2,8–10]. Most studies on phase separation referred to stem cells focus on embryonic stem cells (ESCs) during organ development instead of adult stem cells (ASCs, also known as resident stem cells)[11,12] during tissue homeostasis and aging. A recent finding revealing that phase separation regulates the asymmetric localization of Numb and PAR complexes and contributes to neuroblast division in neural stem cells (NSC), suggests a possible regulatory role of LLPS on

[1]Laboratory for Aging and Cancer Research, Frontiers Science Center Disease-related Molecular Network, State Key Laboratory of Respiratory Health and Multimorbidity and National Clinical Research Center for Geriatrics, West China Hospital, Sichuan University, 610041 Chengdu, Sichuan, China. [2]Laboratory of Metabolism and Aging Research, Frontiers Science Center Disease-related Molecular Network, State Key Laboratory of Respiratory Health and Multimorbidity and National Clinical Research Center for Geriatrics, West China Hospital, Sichuan University, 610041 Chengdu, Sichuan, China. [3]Key Laboratory of Gene Engineering of the Ministry of Education, State Key Laboratory of Biocontrol, School of Life Sciences, Sun Yat-sen University, 510275 Guangzhou, Guangdong, China. [4]Department of Gastroenterology & Hepatology, West China Hospital, Sichuan University, 610041 Chengdu, Sichuan, China. [5]Sichuan University-University of Oxford Huaxi Joint Centre for Gastrointestinal Cancer, Frontiers Science Center for Disease-Related Molecular Network, West China Hospital, Sichuan University, 610041 Chengdu, Sichuan, China. ✉e-mail: chenhy82@scu.edu.cn; haojiang@scu.edu.cn

some ASCs[13,14]. However, no protein has been shown to undergo functionally relevant phase transition during ASCs-mediated tissue regeneration or aging, typically regulated by genetic mutations, epigenetic changes, and extrinsic factors, including glucose, lipid, $Ca^+$ metabolism, and niche environment[15–17]. To evaluate this phenomenon, we utilize intestinal stem cells (ISCs) in the midgut of *Drosophila* as the ASCs model to study tissue regeneration and aging, which divide asymmetrically and differentiate into enteroblasts (EBs) or enteroendocrine mother cells (EMCs) to maintain intestinal homeostasis upon subjected to external injuries or the aging process[18,19].

$N^6$-methyladenine mRNA (m6A) modification is the most prevalent posttranscriptional RNA modification of the common RRACH (R = G or A, H = A, C or U) sequences[20]. The m6A modification modulated by writers (RNA methyltransferases), readers (m6A recognizers), and erasers (RNA demethylases)[20], is required for distinct cellular processes under different cellular contexts[20]. Although previous studies have shown that m6A writer Mettl14 and reader YTHDF1 separately facilitate ISC maintenance and gut homeostasis[21–23], no m6A regulator has been shown to function in ISC aging. Beyond its mitotic functions in microtubule (MT)–kinetochore interaction, spindle assembly checkpoint (SAC), and spindle assembly, BuGZ is also required for proper mRNA splicing, synaptic vesicles (SV) cycling, and hESC reprogramming during interphase[24–26]. BuGZ shows the condensation behavior involved in the MT polymerization during mitosis[27]. Nonetheless, whether BuGZ undergoes phase separation during interphase, and whether BuGZ and its condensation behavior contribute to ISC proliferation and aging are still unknown. In this study, we uncover the role of phase separation on ASC-dependent regeneration and aging, the mechanism for controlling the corresponding TF coacervation via its downstream promotor DNA, and the effect of m6A regulators on modulating the phase transition of the key driver in ASCs.

## Results

### BuGZ forms coacervation puncta in ISCs
BuGZ facilitates the mitosis of cancer cells and enhances self-renewal and pluripotency of human ESCs[24,26,27]. However, its role in ASCs is still unknown. To investigate this phenomenon, a FLAG-tagged BuGZ protein-trap reporter line[28] was used to observe the expression pattern of endogenous BuGZ in *Drosophila* intestines. Immunofluorescence analyses showed that BuGZ was specifically expressed in the nuclei of esg-lacZ[+] ISCs and progenitor cells, Pros[+] EEs (Supplementary Fig. 1a), Dl[+] ISCs (Supplementary Fig. 1b), and NRE-lacZ[+] EBs (Supplementary Fig. 1c), but not in differentiated ECs with Polyploidy nuclei (Supplementary Fig. 1a) in *Drosophila* midguts. Interestingly, BuGZ formed nuclear puncta in the esg[+] ISCs and progenitor cells of aged *Drosophila* (Fig. 1a). The expression level of BuGZ (Fig. 1b), and the size, and number of BuGZ puncta also increased in the esg[+] cells of aged *Drosophila* (Fig. 1c, d). Furthermore, real-time-quantitative polymerase chain reaction (RT-qPCR) (Fig. 1e) and western blotting (Supplementary Fig. 1e) data of the sorted esg[+] cells (Supplementary Fig. 1d) also showed that both mRNA and protein levels of BuGZ were elevated in aged-esg[+] cells. Moreover, bleomycin (BLM)- or paraquat (PQ)-induced gut injury elevated BuGZ levels (Fig. 1e, f), promoting the formation of nuclear puncta in Dl[+] ISCs (Fig. 1g–i) of *Drosophila*.

To further verify that BuGZ undergoes concentration-dependent phase separation in esg[+] cells, we used the *esg-Gal4, UAS-GFP, tub-Gal80[TS]* system (hereafter referred to as esg[TS])[29] to overexpress HA-tagged BuGZ cDNA in esg[+] cells. We found that overexpressing BuGZ indeed induced BuGZ puncta formation in the esg[+] cells of young *Drosophila* (Fig. 1j). To test whether the BuGZ puncta exhibited phase separation property, we exposed the BuGZ-overexpressing *Drosophila* midguts to different temperatures because BuGZ phase separation is sensitive to temperature shifts[27]. We found that BuGZ puncta diffused when the midguts were transferred from 25 to 4 °C, and the diffused protein concentrated again to form more but smaller puncta when

these low temperature-cultured midguts were transferred back to 25 °C with or without MG132 (which was used for inhibiting protein degradation) treatment (Fig. 1j–m and Supplementary Fig. 1f–i). Moreover, the purified His-tagged *Drosophila* BuGZ protein in *E. coli* (Supplementary Fig. 2a) formed droplets in vitro and was regulated by a crowding agent, salt hydronium concentration, temperature and pH (Supplementary Fig. 2b–e), consistent with our previous study[27]. The His-dBuGZ protein also exhibited the fusion property (Supplementary Fig. 2f). All these data indicated that *Drosophila* BuGZ protein indeed underwent liquid-liquid phase separation (LLPS) in vivo and in vitro.

In addition, we found that BuGZ reversible puncta was partially located with TriMethyl-Histone H3-K27-labeled chromatins, but not the Fibrillarin-labeled nucleus, SUMO-labeled Promyelocytic leukemia (PML) bodies, HP1α-labeled heterochromatins, or γH2AvD-labeled DNA repair foci (Fig. 1n–r).

### BuGZ regulates ISC division after injury
The specific localization pattern of BuGZ in the *Drosophila* intestinal stem and progenitor cells suggested that it might regulate intestine regeneration and aging. To verify this hypothesis, we first determined whether BuGZ controlled the proliferation of ISCs. We found that BuGZ depletion with two different RNA interference (RNAi) lines in the esg[+] cells (Supplementary 3a) did not affect the number of esg[+] (ISC and progenitor cell), Delta[+] (Dl; labels ISC cells) or phospho-histone H3[+] (pH3[+]; an immunomarker of cell proliferation) cells in young (7 days) *Drosophila* under normal conditions (Fig. 2a–c and Supplementary 3c, d). However, BuGZ depletion in esg[+] cells diminished ISC proliferation (indicated by increased esg[+] cells, pH3[+] and Dl[+] cells in the midguts) after the BLM-induced gut injury (Fig. 2a–c and Supplementary 3c, d). Thus, we proposed that BuGZ enhanced ISC proliferation to repair the damaged gut epithelial after injury but was not required for ISC basic maintenance.

Furthermore, the cell-type-specific *BuGZ* RNAi in the ISC cells induced by ISC[TS] (NRE-lacZ; esg-GAL4, UAS-GFP/CyO; tub-Gal80ts, NRE-Gal80) inhibited the proliferation of ISC after injury (Fig. 2d). BuGZ depletion in EBs induced by NRE[TS] (NRE-GAL4, UAS-GFP, tub-Gal80ts/CyO), or in EEs by 386y-Gal4 had no effect on the proliferation of ISC following the BLM treatment (Fig. 2e, f). Moreover, different from inhibited BuGZ level in ISCs upon *BuGZ* RNAi using ISC[TS], BuGZ-specific depletion in EBs or EEs did not affect the protein expression of BuGZ in ISCs (Supplementary Fig. 3e–h). Thus, BuGZ mediated the ISC proliferation and gut repair after injury by performing cell-autonomous functions.

To further determine the functions of BuGZ in ISC proliferation, we generated two *BuGZ-null* mutant alleles (BuGZ[3] and BuGZ[8]) using the CRISPR/Cas9 knock-out system (Fig. 2g). We found that *BuGZ-null* mutants of *Drosophila* (BuGZ[3]/BuGZ[8]) showing efficient depletion effect (Supplementary Fig. 3i) died before the white pupae stage. We analyzed the *BuGZ-null* clones via mosaic analysis with a repressible cell marker (MARCM), which labels all progenies of a single active ISC from one clone with a visible GFP marker. After 10 days of clone induction, we found that the *BuGZ-null* clones were smaller than the wild-type clones, and the ISC proliferation defect observed in the *BuGZ-null* clones could be reversed by expressing the full-length BuGZ cDNA (BuGZ[FL]) (Fig. 2h–i). The Flip-out (F/O) lineage-tracing analyses of the RNAi clones also showed that BuGZ depletions suppressed the ISC proliferation after clone induction (Fig. 2j, k).

Previous studies have shown that BuGZ depletion blocked prometaphase-metaphase transition and elevated the mitotic index in a symmetric division of tumor cells[24]. Conversely, we found that *BuGZ* RNAi reduced mitotic index (indicated by decreased esg[+] cells and pH3[+] cells in the midguts) in the asymmetric cell division of ISC after injury. To understand this phenomenon, we evaluated whether BuGZ regulates the asymmetric cell division process of *Drosophila* ISCs by quantifying the proportions of *BuGZ* RNAi or control cells at different

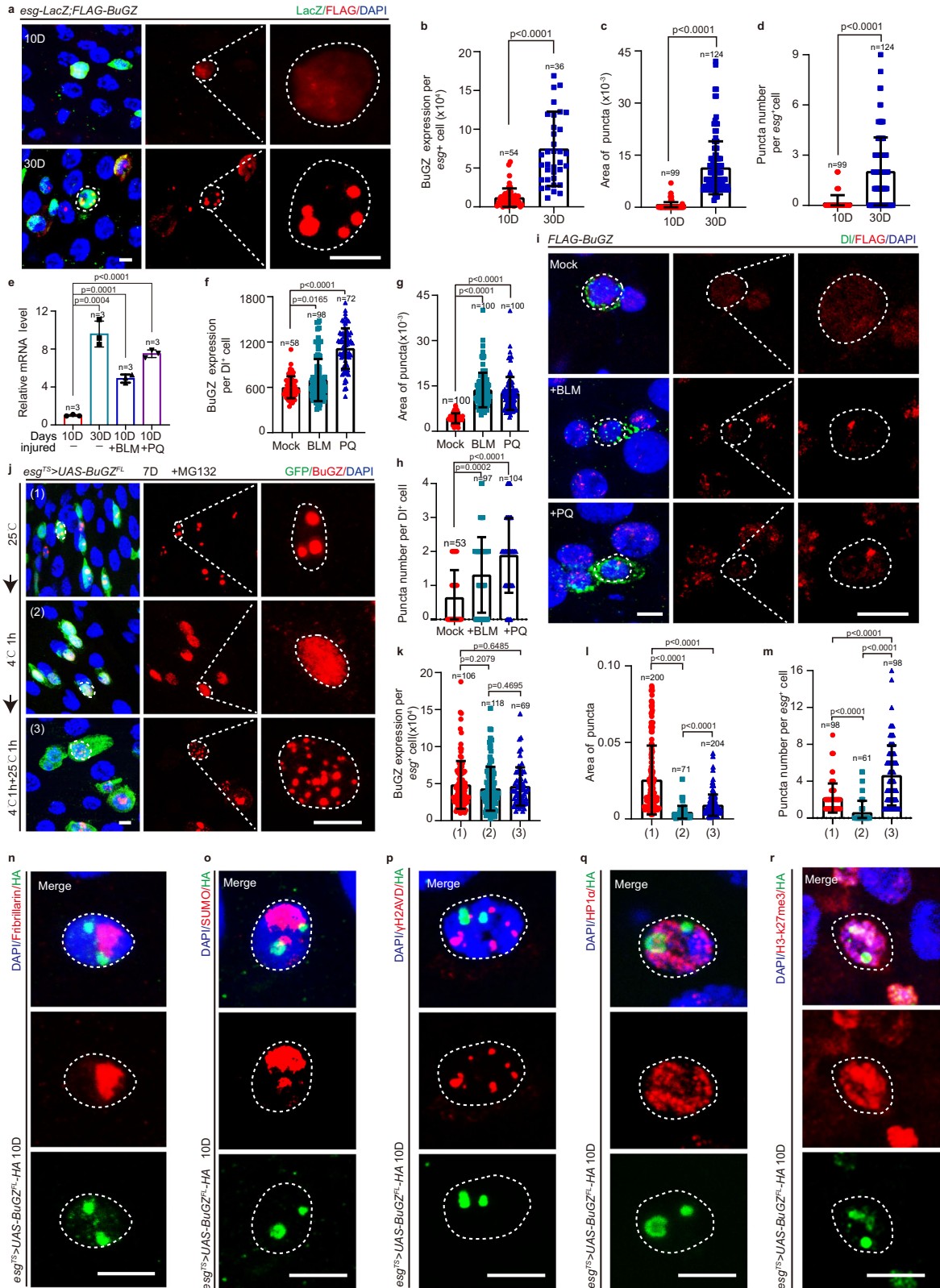

mitotic stages using pH3[+] and DAPI as markers after the BLM-induced injury. The results indicated that the proportion of prophase/prometaphase or metaphase *BuGZ* RNAi cells showed no difference with the control (Supplementary Fig. 3j–l), suggesting that BuGZ had no similar effect on the asymmetric division of ISCs as the symmetric division of tumor cells. Furthermore, the proliferation rate of ISCs (indicated by EdU) in the S phase induced by BuGZ depletion was significantly

reduced after the injury (Supplementary Fig. 3m, n). Thus, BuGZ was not required for the asymmetric division of ISCs but was essential for regulating the proliferation rate during the S phase.

According to the terminal-deoxynucleotidyl transferase-mediated dUTP nick end labeling (TUNEL) assay, *BuGZ-null* clones did not show higher apoptotic signals than control clones (Supplementary Fig. 3o, p). By checking the cell-type-specific makers in the *BuGZ-null* clones, we

**Fig. 1 | BuGZ forms condensation puncta in ISCs during aging and after injury.** **a–d** BuGZ forms puncta in *esg*[+] cells during aging. Image of endogenous BuGZ (red) in *esg*[+] cells (**a**). LacZ indicates *esg*[+] cells (ISCs and EBs, green), nucleus is labeled by DAPI (blue). Fluorescence intensity of BuGZ (**b**). Area (**c**), number (**d**) of BuGZ puncta. Each dot corresponded to one *esg*[+]/BuGZ[+] cell and DAPI is used to calculate the area of the nucleus (**b**, 10D: 54 cells; 30D:36 cells). If the puncta area is bigger than 0.001 pixels, the puncta area is calculated. Otherwise, the area is calculated as zero (**c**, 10D: 99 area; 30D 124 area). **d** 10D: 99 cells; 30D: 124 cells. **e** BuGZ mRNA level increases in old and BLM/PQ-injured *esg*[+] cells (*n* = 3 biologically independent experiments). **f–i** BuGZ forms puncta in BLM-/PQ-injured ISCs. Images of injured BuGZ puncta (**i**). Dl (Delta) labeled ISCs (**i**). Fluorescence intensity (**f**), area (**g**), number (**h**) of BuGZ or BuGZ puncta. (**f**, Mock: 58 cells; BLM: 98 cells; PQ: 72 cells. **g**, 100 areas. **h**, Mock: 53 cells; BLM: 97 cells; PQ: 104 cells). **j–m** BuGZ undergoes concentration- or temperature-dependent phase separation. Image of BuGZ puncta (**j**). BuGZ is labeled by BuGZ antibody (red), GFP indicates *esg*[+] cells (green). Quantification of the fluorescence intensity (**k**), area (**l**), number (**m**) of BuGZ or BuGZ puncta. (**k**, upper: 106 cells; middle: 118 cells; below: 69 cells. **l**, upper: 200 areas; middle: 71 areas; below: 204 areas. **m**, upper: 98 cells; middle: 61 cells; below: 98 cells). **n–r** Puncta of BuGZ does not co-locate with nucleolus marked with Fibrillin (**n**), PML body marked with SUMO (**o**), γH2AVD labeled DNA repair foci (**p**), HP1α labeled heterochromatin (**q**), but partially co-locate with chromatin marked with H3K27me3 (**r**). The experiments were independently repeated three times, yielding similar results. Scale bars represent 5 µm. Bars are mean ± SD. *P*-values were calculated by two-tailed, unpaired Student's *t*-test. Source data are provided as a Source Data file.

found that the presence of Dl[+] ISCs, Pros[+] EEs, NRE-LacZ[+] EBs, and polyploid ECs (Supplementary Fig. 3q–s) in *BuGZ-null* clones on day10 after clone induction (ACI). This proposed that BuGZ was indispensable for ISC proliferation and might not be important for ISC differentiation.

## BuGZ leads to gut hyperplasia upon aging
As *Drosophila* ages, the ISCs in their midguts undergo uncontrolled hyperproliferation even without injury[18,19]. Since BuGZ level was elevated in ISCs of aged *Drosophila* (Fig. 1b, e and Supplementary Fig. 1e), we wanted to know whether increasing BuGZ level contributed to age-associated ISC hyperproliferation. We found that overexpressing *BuGZ*[FL] in the *esg*[+] cells of young *Drosophila* elevated the ISC proliferation rate (Fig. 2l–n and Supplementary Fig. 3b), and BuGZ[FL]-overexpressing MARCM clones were also bigger than the control clones (Fig. 2h, i), indicating that BuGZ overexpression led to an increase in ISC proliferation. Additionally, we found that BuGZ depletion in ISCs of aged *Drosophila* induced by *esg*[TS]-driven RNAi significantly rescued the age-associated ISC hyperproliferation phenotype (Fig. 2l–n). Based on these data, it could be inferred that age-associated BuGZ elevation promoted ISC proliferation upon aging.

## LLPS of BuGZ mediates ISC proliferation
Since BuGZ augmentation resulted in condensation and increased proliferation of ISCs, we examined whether the ISC proliferation was controlled by BuGZ coacervation in aged or damaged midguts. We generated a UAS transgenic fly expressing a modified BuGZ[13S] protein whose 6 tyrosines (Y) and 7 phenylalanines (F) were replaced by serines to disrupt the phase separation property of BuGZ, as previously reported[27] (Fig. 3a). Overexpression of BuGZ[13S] under *esg*[TS] diminished BuGZ puncta formation compared with BuGZ[FL] overexpression under the same intensity of protein level (Fig. 3b–e). BuGZ[13S] overexpression in *esg*[+] cells showed fewer *esg*[+] and pH3[+] cells, compared with BuGZ[FL] overexpression (Fig. 3b, f, g). BuGZ[13S]-overexpressing MARCM clones also had smaller sizes than BuGZ[FL]-overexpressing clones (Fig. 3h, i). Meanwhile, cell number reduction in *BuGZ-null* clones was prevented by overexpressing BuGZ[FL] but not overexpressing BuGZ[13S] (Fig. 3h, i). Thus, it revealed that the ability of BuGZ protein to form phase-separation-mediated puncta was essential for its function in ISC proliferation.

The presence of intrinsically disordered regions (IDRs), modulating the emergent behavior and properties of LLPS, is one of the common molecular signatures of phase separation[3,5,6]. Our previous study showed that BuGZ peptide C (BuGZ[C]) in the IDR region, but not CS (BuGZ[CS]) (Fig. 3a) suppressed the coacervation of wild-type BuGZ in vitro[27]. We found that overexpressing either BuGZ[C] or BuGZ[CS] in *esg*[+] cells in young *Drosophila* had no significant changes in ISC proliferation under normal conditions (Fig. 3j–l). Interestingly, BuGZ[C] overexpression in *esg*[+] cells inhibited ISC proliferation, whereas BuGZ[CS] overexpression showed less effect than BuGZ[C] on ISC proliferation after BLM-induced injury (Fig. 3j–l). Moreover, compared with BuGZ[CS], BuGZ[C] overexpression shows more effective suppression for BuGZ condensate in ISCs caused by gut repair (Fig. 3m). To further determine the effects of BuGZ[C] and BuGZ[CS] peptides on regulating ISC proliferation, we co-expressed BuGZ[FL] with BuGZ[C] or BuGZ[CS] in *esg*[+] cells. We observed that BuGZ[C] showed stronger effects than BuGZ[CS] on reducing the size of BuGZ puncta resulting from BuGZ overexpression and activation of ISC proliferation (Fig. 3n–r). Thus, the truncated IDR peptide BuGZ[C] inhibited ISC proliferation by disrupting the phase separation of BuGZ, and BuGZ[CS] attenuated the peptide interference of BuGZ coacervation and ISC proliferation, which confirmed that BuGZ regulated the ISC proliferation based on its phase separation property.

## BuGZ affects gut repair and longevity
Previous studies have reported that the misregulation of ISCs leads to a significant decrease in digestive functions in aging *Drosophila*[19,30]. This, in turn, reduces the lifespan of *Drosophila*[31]. Since BuGZ level increased in ISCs upon aging, we explored whether BuGZ elevation and its phase separation promoted *Drosophila* intestine aging.

As expected, overexpressing BuGZ[FL] in ISCs reduced the food intake (Supplementary Fig. 4a, c), disrupted the gastrointestinal acid-base homeostasis (Supplementary Fig. 4a, d), and inhibited excretion (Supplementary Fig. 4b, e) in young flies. However, these defects in *Drosophila* overexpressing BuGZ[13S] (which shows diminished property of BuGZ LLPS) were not as severe as in *Drosophila* overexpressing BuGZ[FL] (Supplementary Fig. 4a–e). Similar to the effects of BuGZ[C] or BuGZ[CS] on BuGZ phase separation and ISC proliferation (Fig. 3m–r), the effect of BuGZ[C] on alleviating gastrointestinal acid-base homeostatic disruption, reduction of food intake, and suppression of excretion was stronger in young overexpressing BuGZ[FL] flies or in aged flies than those overexpressing BuGZ[CS] (Supplementary Fig. 4a–h). Furthermore, BuGZ depletion significantly reversed the decline of intestinal digestive functions in aged *Drosophila* (Supplementary Fig. 4a, b, f–h).

Next, we analyzed whether BuGZ or its phase separation regulated *Drosophila* lifespan. *BuGZ* RNAi in the *esg*[+] cells extended the lifespan of *Drosophila* after eclosion under normal conditions (Supplementary Fig. 4i). Overexpression of BuGZ[FL] in ISCs effectively shortened *Drosophila* longevity, whereas BuGZ[13S] overexpression had fewer effects on reducing the *Drosophila* lifespan (Supplementary Fig. 4i). Additionally, overexpressing BuGZ[C] peptides prolonged the lifespan of *Drosophila* with or without co-expressing BuGZ[FL] more significantly than BuGZ[CS] (Supplementary Fig. 4j).

Since the integrity of the epithelial barrier and length of the midgut were associated with gut turnover[32,33], we assessed the integrity and length of the intestine in BuGZ[C/CS] overexpressed ISCs. Armadillo immunostaining assay, which was used to test the integrity of gut epithelium[33], indicated that overexpression of BuGZ[C] induced unrecoverable injury to the intestinal epithelium (Supplementary Fig. 4k), which also resulted in shortened length of the midguts (Supplementary Fig. 4l, m). Additionally, consistent with *BuGZ* RNAi (Supplementary Fig. 4n), overexpression of BuGZ[C] caused higher mortality following BLM-induced chronic injury (Supplementary Fig. 4o).

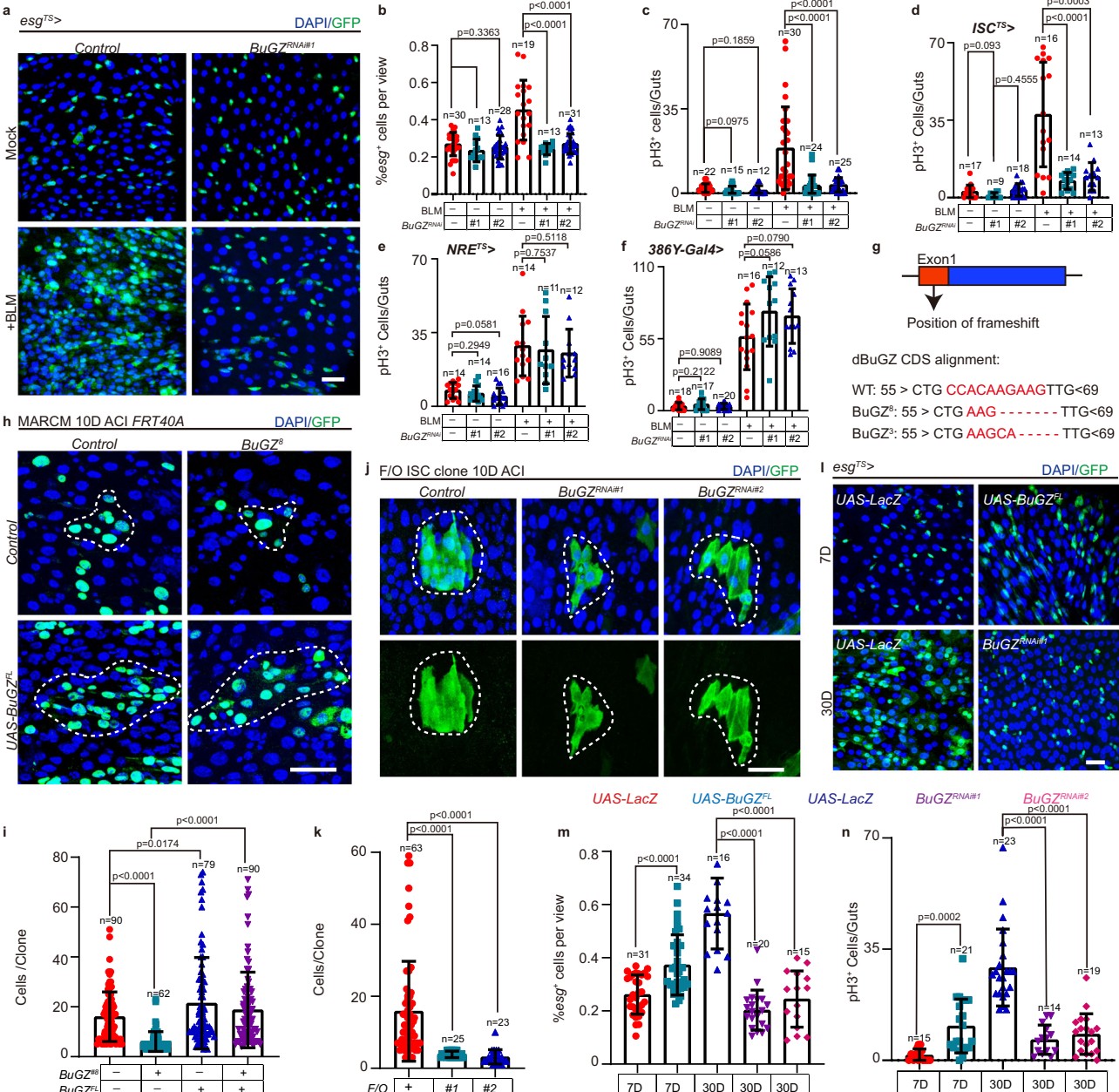

**Fig. 2 | BuGZ regulates activation of ISC proliferation after gut injury and during aging. a–c** BuGZ controls ISC proliferation after gut injury. Images of *esg*[+] cells in control (*UAS-LacZ*), *BuGZ*[RNAi#1] (BDSC:27996), *BuGZ*[RNAi#2] (VDRC: v104498) midguts (**a**). Ratio of *esg*[+] cells to DAPI cells (**b**, from left to right are 30, 13, 28, 19, 13, 31 views). pH3[+] cells per midguts (**c**, from left to right are 22, 15, 12, 30, 24, 25 guts). GFP indicates *esg*[+] cells (green). **d–f** BuGZ mediates ISC proliferation after injury by performing cell-autonomous functions. pH3[+] cells per young (7D) or injured midguts under *ISC*[TS] (**d**), *NRE*[TS] (**e**), *386Y-gal4* (**f**). (**d**: from left to right are 17, 9, 18, 16, 14, 13 guts; **e**: from left to right are 14, 14, 16, 14, 11, 12 guts; **f**: from left to right are 18, 17, 20, 16, 12, 13 guts). **g** Model of mutually independent *BuGZ-null* allele. **h**, **i** BuGZ depletion suppresses ISC proliferation after clone induction. Images of MARCM clones (**h**, green, outlined by white dotted lines). Number of cells per clone (**i**, from left to right are 90, 62, 79, 90 clones). GFP indicates ISCs (green). **j**, **k** Flip-out(F/O) lineage-tracing clone system is used to mark ISCs and their newborn progenies. Images of F/O clones (**j**). Number of cells per clone (**k**, from left to right are 63, 25, 23 clones). GFP indicates ISCs (green). **l–n** BuGZ elevation leads to an increase ISC proliferation upon aging. Images of midguts with *esg*[+] staining (**l**). GFP indicates *esg*[+] cells (green). Ratio of *esg*[+] cells to DAPI cells (**m**, from left to right are 31, 34, 16, 20, 15 views). Number of pH3[+] cells per midgut (**n**, from left to right are 15, 21, 23, 14, 19 views). DAPI stained nuclei (blue). Scale bars represent 25 μm. View size is $3.4 \times 10^4$ μM[2]. Bars are mean ± SD. *P*-values were calculated by two-tailed, unpaired Student's *t*-test. Source data are provided as a Source Data file.

Besides, BuGZ[CS] showed compromised effects on the intestinal integrity, length and lifespan when compared with BuGZ[C] (Supplementary Fig. 4k–m and Supplementary Fig. 4o). Based on these findings, we concluded that BuGZ phase separation played a crucial role on promoting intestinal integrity during gut repair.

Thus, we showed that BuGZ and its phase separation affected the lifespan and gut repair of *Drosophila*, mediated by gut homeostasis (Supplementary Fig. 4p).

## BuGZ promotes ISC division by YT521-B

Previous studies showed a strong relationship between RNA-associated proteins and BuGZ[34,35], we noticed that BuGZ showed strong interactions of BuGZ with RNA-associated proteins, especially with m6A-related proteins in ESCs, inferring that m6A-related proteins might be involved in the physiological functions of BuGZ in stem cells (Supplementary Fig. 5a). To investigate the underlying mechanism by which BuGZ regulates the proliferation of ISC, we

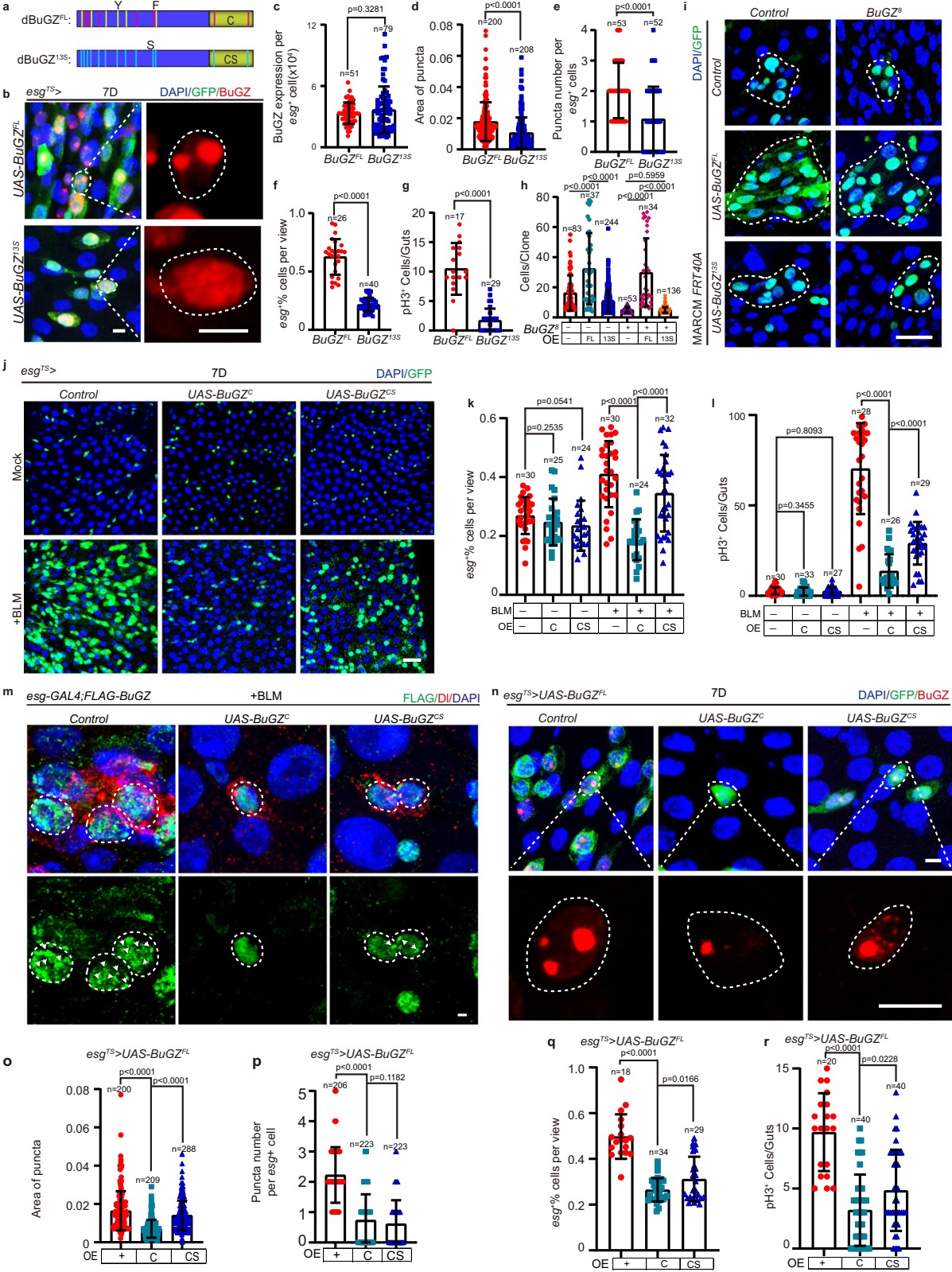

performed a genetic screen with m⁶A-associated genes to uncover the potential genes that could rescue *BuGZ* RNAi phenotype in regulating the ISC proliferation after gut injury (Supplementary Fig. 5b). We found that *YT521-B* (m⁶A reader) RNAi rescued the suppressed ISC proliferation in BuGZ depletion *Drosophila* after BLM-induced injury (Fig. 4a–c). Furthermore, inhibiting YT521-B expression restored

the disrupted ISC proliferation in *BuGZ-null* MARCM clones (Fig. 4d, e), and overexpression of YT521-B diminished ISC hyperproliferation caused by BuGZ overexpression (Fig. 4f–h). Both depletion and overexpression of YT521-B did affect its expression level (Supplementary Fig. 5c). Consistently, depletion of YT521-B in ISCs using *esg^TS* increased pH3⁺ cells and *esg⁺* cells in the midguts of young or

**Fig. 3 | LLPS of BuGZ regulates ISC proliferation. a** Model of BuGZ mutants (BuGZ^13S^, BuGZ^C^, and BuGZ^CS^). **b–g** ISC proliferation is controlled by BuGZ coacervation. Images of midguts with *esg*^+^/BuGZ^+^ staining (**b**). BuGZ is indicated by the BuGZ antibody. GFP indicates *esg*^+^ cells (green). Fluorescence intensity (**c**), area (**d**), number (**e**) of BuGZ puncta. (**c**, *UAS-BuGZ^FL^*, *UAS-BuGZ^13S^*: 51, 79 cells. **d**, *UAS-BuGZ^FL^*, *UAS-BuGZ^13S^*: 200, 208 cells. **e**, *UAS-BuGZ^FL^*, *UAS-BuGZ^13S^*: 53, 52 cells). Ratio of *esg*^+^ to DAPI cells (**f**, *UAS-BuGZ^FL^*, *UAS-BuGZ^13S^*: 26, 40 views). pH3^+^ numbers per midgut (**g**, *UAS-BuGZ^FL^*, *UAS-BuGZ^13S^*: 17, 29 guts). **h, i** LLPS of BuGZ is essential for ISC proliferation. Images of MARCM clones (**i**). Cell numbers per clone (**h** from left to right: 83, 37, 244, 53, 34, 136 clones). GFP indicates ISCs (green). **j–l** BuGZ phase separation regulates ISC proliferation upon gut injury. Images of *esg*^+^ cells (**j**). Ratio of *esg*^+^ to DAPI cells (**k**, from left to right: 30, 25, 24, 30, 24, 32 views). pH3^+^ numbers per midgut (**l**, from left to right: 30, 33, 27, 28, 26, 29 guts). GFP indicates *esg*^+^ cells

(green). **m** Overexpression of BuGZ^C^ inhibited the LLPS of BuGZ during gut repair, while BuGZ^C^ exhibited more effective influence than BuGZ^CS^ on suppressing BuGZ condensate. Dl labels ISCs. White arrow indicates puncta. Experiments were independently repeated three times. **n–r** Truncated IDR peptide BuGZ^C^ instead of BuGZ^CS^ inhibits ISC proliferation by disrupting LLPS of BuGZ. Images of midguts with *esg*^+^/BuGZ^+^ staining. BuGZ is indicated by BuGZ antibody (**n**). GFP indicates *esg*^+^ cells (green). Quantification of the area (**o**), number (**p**) of BuGZ puncta (**o**, from left to right: 200, 209, 288 areas. **p**, from left to right: 206, 223, 223 cells). Ratio of *esg*^+^ to DAPI cells (**q**, from left to right: 18, 34, 29 views). pH3^+^ numbers per midgut (**r**, 20, 40, 40 guts). Scale bars represent 5 μm (**b, n**), 10 μm (**m**), 25 μm (**i, j**). View size is 3.4 × 10⁴ μM². Bars are mean ± SD. *P*-values were calculated by two-tailed, unpaired Student's *t*-test. Source data are provided as a Source Data file.

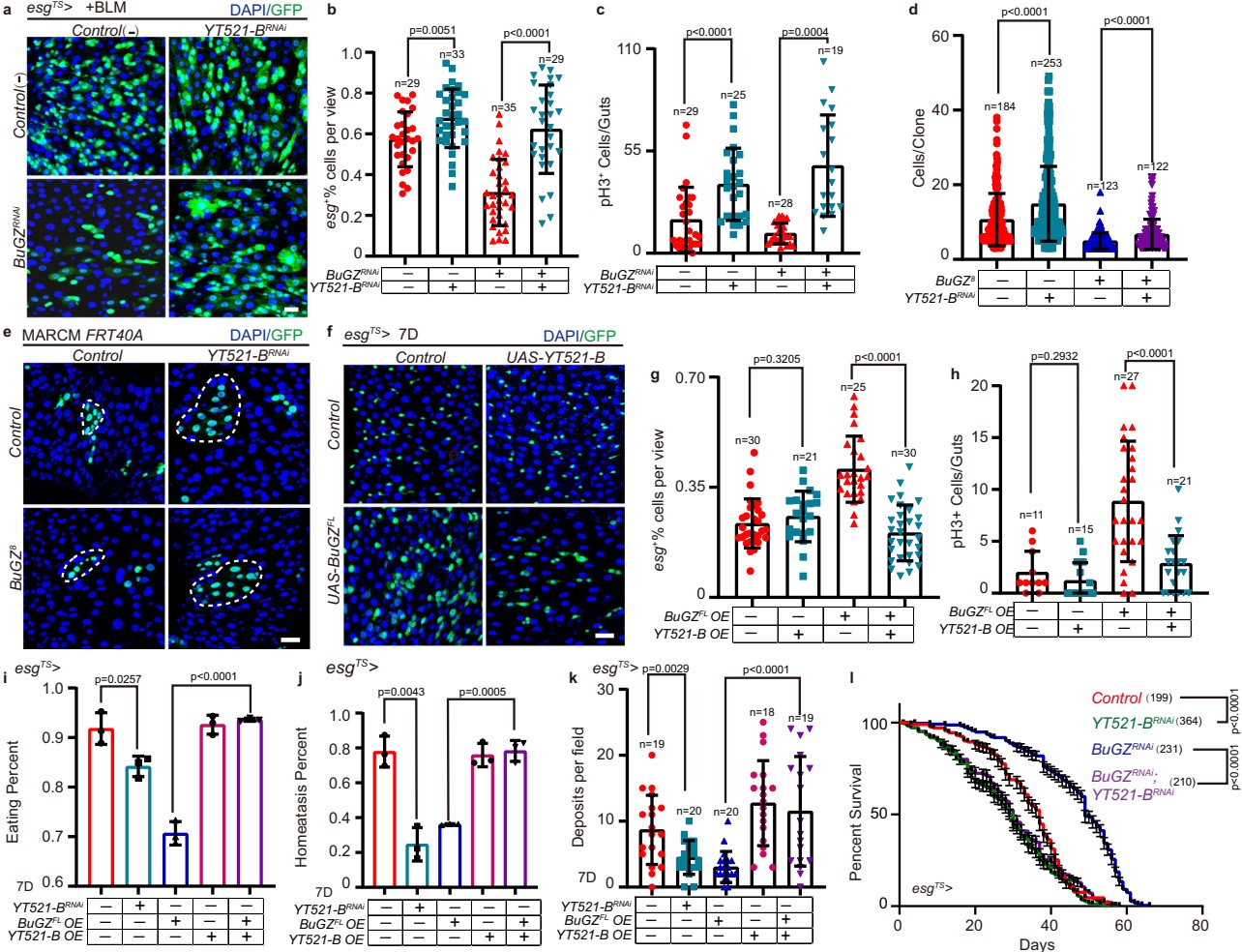

**Fig. 4 | BuGZ facilitates *Drosophila* ISC proliferation via m⁶A reader YT521-B. a–c** Expression of *YT521-B^RNAi^* promotes the proliferation of ISCs and rescues the defect of ISCs proliferation caused by BuGZ depletion after injury in *esg*^+^ cells. Immunofluorescence images of *esg*^+^ cells (**a**). Ratio of *esg*^+^ cells to DAPI cells (**b**, from left to right are 29, 33, 35, 29 views). Number of pH3^+^ cells per midgut (**c**, from left to right are 29, 25, 28, 19 guts). GFP indicates *esg*^+^ cells (green). Bars are mean ± SD, Two-tailed unpaired Student's *t*-test. **d, e** Immunofluorescence images of MARCM clones (green, outlined by white dotted lines) (**d**). Cell numbers per clone (**e**, from left to right are 184, 253, 123, 122 clones, bars are mean ± SD, Two-tailed unpaired Student's *t*-test.). GFP indicates ISCs (green). **f–h** Overexpression of YT521-B significantly restrains the hyperproliferation of ISCs caused by BuGZ^FL^-overexpressing. Representative images of *esg*^+^ cells (**f**). Ratio of *esg*^+^ cells to DAPI cells (**g**, from left to right are 30, 21, 25, 30 views). Number of pH3^+^ cells per midgut (**h**, from left to right are 11, 15, 27, 21 guts). GFP indicates *esg*^+^ cells (green). Bars are

mean ± SD, Two-tailed unpaired Student's *t*-test. **i** Percentage of the ratio of flies of eating to each indicated genotypes of young (7D) flies. Error bars show the SD of three independent experiments. Two-tailed unpaired Student's *t*-test. **j** Percentage of the intestinal homeostasis categories from flies with indicated genotypes. Error bars show the SD of three independent experiments. Two-tailed unpaired Student's *t*-test. **k** Excretion of *Drosophila* treated with Bromophenol blue from young (7D) flies with indicated genotypes. Tests were repeated as three independent experiments. Two-tailed unpaired Student's *t*-test. **l** Percentage of survival of adults under *esg*^TS^. (*Control*: *n* = 199 flies; *YT521-B^RNAi^*: *n* = 364 flies; *BuGZ^RNAi^*: *n* = 231 flies; *BuGZ^RNAi^*; *YT521-B^RNAi^*: *n* = 210 flies). Tests were repeated as three independent experiments. Statistical significance among genotypes was calculated with a chi-square log-rank test. Scale bars represent 25 μm. The area of the view size is 3.4 × 10⁴ μM². Source data are provided as a Source Data file.

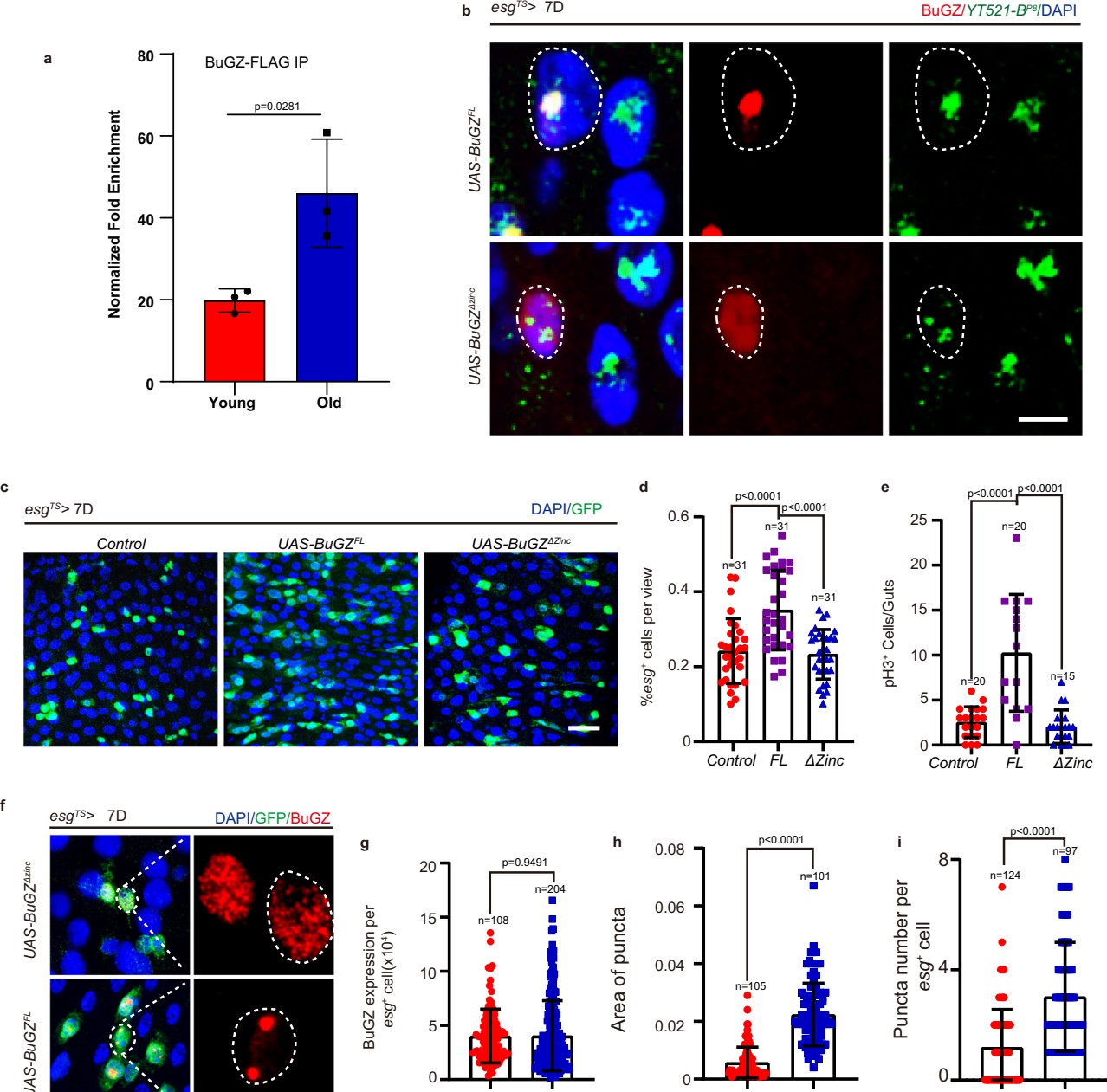

**Fig. 5 | The binding of YT521-B promotor to BuGZ accelerates its coacervation to control the proliferation of ISCs. a** The interaction between BuGZ and $P^8$ fragment by Chip-qPCR in the young or old guts of transgenic BuGZ-FLAG fly. The results demonstrated that the interaction between BuGZ and $P^8$ was enhanced upon aging. **b** The in situ hybridization assay shows that, compared with BuGZ$^{ΔZinc}$, BuGZ$^{FL}$ protein exerted stronger behavior of condensation co-located with the DNA fragment *P8*. The experiments were independently repeated three times, yielding similar results. **c–e** BuGZ$^{ΔZinc}$ does not affect the proliferation of ISCs. Immunofluorescence images of control (*UAS-LacZ*), BuGZ$^{FL}$-HA overexpressing, *BuGZ$^{ΔZinc}$-HA* overexpressing *esg$^+$* cells (**c**). Ratio of *esg$^+$* cells to DAPI cells (**d**, 31 views). Number of pH3$^+$ cells per midguts (**e**, *Control*: n = 20 guts; *UAS-BuGZ$^{FL}$*: n = 20 guts; *UAS-*

*BuGZ$^{ΔZinc}$*: n = 15 guts). GFP indicates *esg$^+$* cells (green). **f–h** Overexpression of BuGZ$^{ΔZinc}$ shows smaller puncta than BuGZ$^{FL}$-overexpressing in *esg$^+$* cells. BuGZ foci (red, outlined by white dotted lines) is labeled by BuGZ antibody (**f**). Quantification of the fluorescence intensity (**g**), area (**h**), number (**i**) of BuGZ puncta. (**g**, *UAS-BuGZ$^{ΔZinc}$*: n = 108 guts; *UAS-BuGZ$^{FL}$*: n = 204 cells. **h**, *UAS-BuGZ$^{ΔZinc}$*: n = 105 areas; *UAS-BuGZ$^{FL}$*: n = 101 areas. **i**, *UAS-BuGZ$^{ΔZinc}$*: n = 124 cells; *UAS-BuGZ$^{FL}$*: n = 97 cells). GFP indicates *esg$^+$* cells (green). DAPI stained nuclei (blue). Scale bars represent 5 μm (**b**, **c**), 25 μm (**f**). The area of the view size is $3.4 × 10^4$ μM$^2$. Bars are mean ± SD. *P*-values were calculated by two-tailed, unpaired Student's *t*-test. Source data are provided as a Source Data file.

injured *Drosophila* (Supplementary Fig. 5d–f and Fig. 4a–c). YT521-B-depleted MARCM clones were bigger than the control clones (Fig. 4d, e). Overexpression of YT521-B had no effect on regulating the proliferation of ISC in young *Drosophila* (Fig. 4f–h) but inhibited ISC hyperproliferation in old or injured *Drosophila* (Supplementary Fig. 5g–i).

In addition, inhibiting YT521-B expression disrupted the intestinal digestive functions in young *Drosophila*, while YT521-B overexpression rescued the intestinal digestion defects caused by the overexpression of BuGZ in ISCs (Fig. 4i–k). Moreover, *YT521-B* RNAi of the ISCs decreased the lifespan and augmented the mortality of BuGZ-depleted *Drosophila* with aging (Fig. 4l). Thus, YT521-B acted as the downstream

regulator of BuGZ to regulate ISC proliferation, intestinal function, and longevity of *Drosophila*.

## BuGZ prevents YT521-B transcription

Since BuGZ was shown to function in nuclei as a transcription factor[36], we explored whether BuGZ could affect the transcription of YT521-B. RT-qPCR analyses using sorted *esg⁺* cells showed that the mRNA level of YT521-B increased in BuGZ-depleted *esg⁺* cells (Supplementary Fig. 6a). Western blotting analyses further indicated that inhibiting BuGZ in ISCs upregulated the YT521-B protein level (Supplementary Fig. 6b). Accordingly, the mRNA and protein levels of YT521-B in ISCs decreased in aged *Drosophila*, with the increase of BuGZ protein level (Supplementary Fig. 6c, d). To verify the roles of BuGZ and its phase transition in controlling the transcription of YT521-B, we first observed the *YT521-B* mRNA level in BuGZ^FL and BuGZ^13S overexpressed ISCs. The results demonstrated that the overexpression of BuGZ^FL, in contrast to BuGZ^13S, inhibited the mRNA level of YT521-B (Supplementary Fig. 6e). Besides, BuGZ^C overexpression increased the *YT521-B* mRNA level upon gut injury, whereas BuGZ^CS overexpression showed compromised effect on prompting YT521-B expression level in relative to BuGZ^C (Supplementary Fig. 6f). Moreover, we generated a transgenic fly (*YT521-B-Luciferase*), carrying a luciferase sequence with a promotor region of YT521-B, to sort out the *esg⁺* cells to perform the following experiments (Supplementary Fig. 6g). We found that *BuGZ* RNAi of ISCs in aged *Drosophila* increased the transcription activity of the YT521-B promotor (Supplementary Fig. 6h). Conversely, overexpression of BuGZ^FL but not BuGZ^13S suppressed the transcription activity of the YT521-B promotor in ISCs of young *Drosophila* (Supplementary Fig. 6h). Additionally, our Chip-qPCR data indicated that BuGZ^FL exhibited enhanced binding to the YT521-B promoter in contrast to BuGZ^13S (Supplementary Fig. 6i). Meanwhile, BuGZ^C overexpression suppressed the binding of the YT521-B promoter to BuGZ in Flag-BuGZ knock-in flies under BLM treatment, whereas BuGZ^CS overexpression displayed diminished effect on attenuating the binding of BuGZ to the YT521-B promoter when compared with BuGZ^C (Supplementary Fig. 6j). Thus, these findings revealed BuGZ condensate inhibits YT521-B transcriptional by promoting DNA binding.

## BuGZ-YT521-B axis regulates MAPK pathway

To identify the downstream pathway by which the BuGZ-YT521-B axis regulated ISC proliferation, RNA-sequencing analyses were performed using *Drosophila* midguts overexpressing BuGZ in *esg⁺* cells. The mitogen-activated protein kinase (MAPK) signaling pathway was detected using the Kyoto Encyclopedia of Genes and Genomes (KEGG) pathway enrichment analysis (Supplementary Fig. 7a), which was reported previously to regulate ISC proliferation and self-renewal[37,38]. Multiple typical regulators that positively controlled the MAPK signaling pathway and ISC proliferation were upregulated in *Drosophila* overexpressing BuGZ (Supplementary Fig. 7b). We found that active MAPK signaling by overexpressing activated versions of EGFR (*UAS-EGFR^CA*, activated form of epidermal growth factor receptor) or Ras85D (*UAS-Ras85D*, Ras oncogene at 85D) accelerated the disrupted ISC proliferation in *BuGZ-null* MARCM clones (Supplementary Fig. 7c, d) and MAPK signaling inhibition by expressing a dominant-negative version of EGFR (*UAS-EGFR^DN*) prevented the ISC hyperproliferation caused by YT521-B depletion (Supplementary Fig. 7e–g). To study the detailed mechanism of how YT521-B modulated the MAPK pathway, we examined the ERK activity assessed by immunostaining against the double phosphorylated ERK kinase (dpERK), which behaved as the key downstream target of the MAPK pathway[39]. Our data showed that YT521-B overexpression inhibited the ERK activation caused by BuGZ overexpression (Supplementary Fig. 7h). BuGZ depletion in ISCs resulted in a significant decrease of dpERK signal in ISCs (Supplementary Fig. 7i), while *YT521-B* RNAi restored the down-regulation of dpERK in ISCs caused by BuGZ depletion (Supplementary Fig. 7i). We

also investigated the impact of BuGZ and YT521-B on the mRNA expressions of *Argos* (*aos*), *Kekkon 1* (*kek1*), and *Sprouty* (*sty*) in sorted *esg⁺* cells, all of which were reported to act as downstream targets of MAPK signaling pathway[37,40–42]. We found that the overexpression of BuGZ^FL augmented the mRNA levels of kek1, sty, and aos (Supplementary Fig. 7j). Accordingly, the overexpression of YT521-B reversed the increased mRNA levels of *kek1*, *sty*, and *aos* induced by BuGZ^FL overexpression (Supplementary Fig. 7j). Moreover, *YT521-B* RNAi rescued the reduction of *aos*, *kek1*, and *sty* mRNA levels caused by *BuGZ* RNAi during gut regeneration (Supplementary Fig. 7k). Thus, BuGZ-YT521-B functioned as the upstream regulator of the MAPK signaling pathway to modulate ISC proliferation in *Drosophila* (Supplementary Fig. 7l).

## YT521-B promotor enhances LLPS of BuGZ

Recent studies have suggested that the 'transcriptional condensates' contain TFs, cofactors, and RNA Pol II, assembled at the enhancer and promoter regions of key cell-identity genes[8]. However, it is still unknown whether the promotor can promote the phase separation of its binding transcription factor and whether the promotor condensates function in cell or organism development. To address this issue, we added three different fractions of the YT521-B promotor regions (Supplementary Fig. 8a) into the BuGZ droplets in vitro. We found that three fragments upstream of the YT521-B gene, especially the first 1 kb fraction (1K¹), enhanced the formation of BuGZ droplets (Supplementary Fig. 8b, c). We also identified a 100 bp DNA fragment within 1K¹ (*P⁸*) that strongly promoted BuGZ phase separation (Supplementary Fig. 8a, d, g, h). Moreover, the Chip-qPCR experiment indicated that BuGZ directly interacted with *P⁸* in young or old flies, and the interaction between BuGZ and *P⁸* was enhanced upon aging (Fig. 5a). To further verify the role of *P⁸* in BuGZ phase separation, we performed in vitro purification of the *Drosophila* BuGZ^ΔZinc domain (with depleted the zinc finger), which was unable to bind DNA in our previous study[27] (Supplementary Fig. 8e). The droplet formation of BuGZ^ΔZinc relied on its protein concentration but not the DNA fragment *P⁸* (Supplementary Fig. 8f–h). Additionally, we generated a transgenic fly, *BuGZ^ΔZinc*, to further elucidate the function of the DNA binding property of BuGZ in its phase separation and proliferation of ISCs. The in situ hybridization assay showed that, unlike BuGZ^ΔZinc, BuGZ^FL protein specifically co-localized with the DNA fragment *P⁸* (Fig. 5b). BuGZ^ΔZinc overexpression in ISCs of young *Drosophila* showed less induction of ISC proliferation and attenuated puncta formation compared with BuGZ^FL overexpression (Fig. 5c–i). To further explore the function of the *P⁸* to BuGZ phase separation, we performed the FRAP assay to observe the dynamics of BuGZ droplets with or without adding *P⁸*. The fluorescence recovery rate after photobleaching showed the DNA fragment *P⁸* inhibited the dynamics of BuGZ condensate (Supplementary Fig. 8i–l). Thus, beyond the simple physical binding, the YT521-B promotor might recruit the BuGZ condensate to stably bind to the transcription region to constantly exert its transcriptional function.

## Ime4 and Mettl14 suppress LLPS of BuGZ

Among the candidate genes that could abrogate the inhibition of ISC proliferation caused by BuGZ depletion after gut injury, beyond YT521-B, the depletions of another two m⁶A regulators Ime4 and Mettl14 also restored the reduction of ISC number with *BuGZ* RNAi (Supplementary Fig. 5a, b and Supplementary Fig. 9a–e). The accumulation of *esg⁻* cells and pH3+ cells in the midguts of young or injured *Drosophila* with either *Ime4* or *Mettl14* RNAi in ISCs showed that both Ime4 and Mettl14 inhibited ISC proliferation (Supplementary Fig. 9c–h). MARCM analysis further demonstrated that either *Ime4* or *Mettl14* RNAi rescued the suppressed ISC proliferation in *BuGZ-null* MARCM clones (Supplementary Fig. 9i, j). *Ime4* or *Mettl14* RNAi MARCM clones grew bigger than the control, further manifesting the effect of Ime4 or Mettl14 on

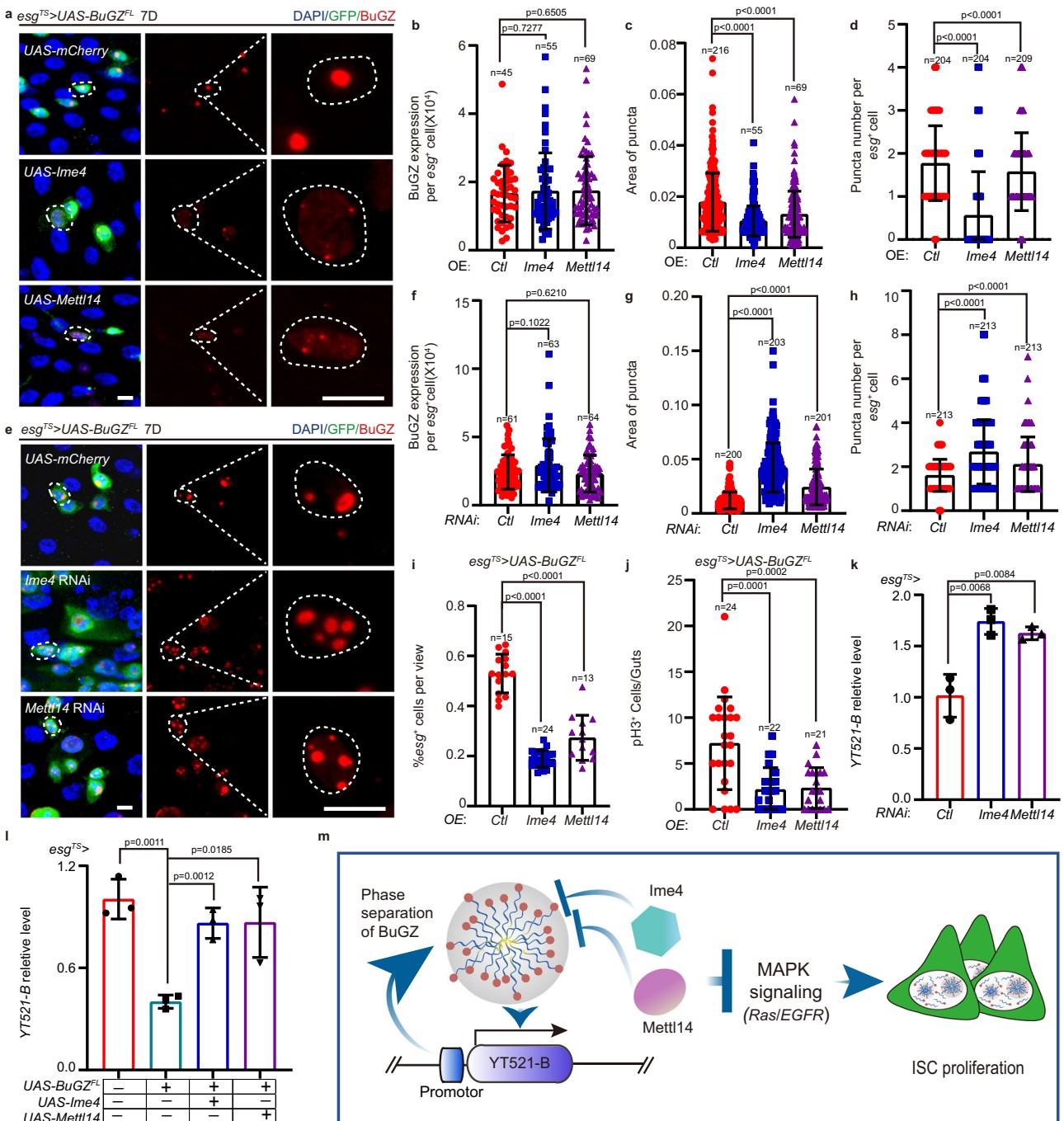

**Fig. 6 | Ime4 and Mettl14 suppress ISC proliferation by inhibiting LLPS of BuGZ.**
**a**–**c** BuGZ foci is inhibited by *Ime4* or *Mettl14* overexpression. Images of *esg⁺* cells
(**a**). BuGZ foci labeled by BuGZ antibody. Fluorescence intensity (**b**), size (**c**),
number (**d**) of BuGZ puncta. (**b**, from left to right are 45 cells, 55 cells, 69 cells.
**c**, from left to right are 216, 55, 69 cells. **d**, from left to right are 204, 204, 209 cells).
GFP indicates *esg⁺* cells (green). **e**–**h** LLPS of BuGZ is promoted by Ime4 or Mettl14
depletion. Images of *esg⁺* cells (**e**). BuGZ foci are labeled by BuGZ antibody.
Fluorescence intensity (**f**), size (**g**), number (**h**) of BuGZ puncta (**f**, from left to right
are 61, 63, 64 cells. **g**, from left to right are 200, 203, 201 puncta. **h**, from left to right
are 213, 213, 213 cells). GFP indicates *esg⁺* cells (green). **i**, **j** Hyperproliferation of ISCs
caused by BuGZ^FL overexpression is inhibited by overexpressing Ime4 and Mettl14
(**h**). Ratio of *esg⁺* cells to DAPI cells (**i**, from left to right are 15, 24, 13 views). Number

of pH3⁺ cells per midgut (**j**, from left to right are 24, 22, 21 guts). **k** Ime4/Mettl14
depletion results in elevated YT521-B mRNA levels in sorted *esg⁺* cells (*n* = 3 biolo-
gically independent experiments). **l** Ime4 or Mettl14 overexpressing rescues YT521-
B transcription suppression caused by BuGZ overexpression in sorted *esg⁺* cells
(*n* = 3 biologically independent experiments). **m** Model of mechanism: how BuGZ
modulates ISC proliferation. LLPS of BuGZ inhibits YT521-B transcription to pro-
mote ISC proliferation by MAPK pathway. Additionally, both Ime4 and Mettl14
regulate YT521-B transcription by inhibiting BuGZ phase separation in ISCs, while in
turn the promotor of YT521-B promotes the LLPS of BuGZ.DAPI stained nuclei
(blue). Scale bars represent 5 μm. Bars are mean ± SD. *P*-values were calculated by
two-tailed, unpaired Student's *t*-test. Source data are provided as a Source Data file.

suppressing the proliferation of ISC (Supplementary Fig. 9i, j). Over-expression of either Ime4 or Mettl14 decreased the ISC proliferation in old and injured, but not young *Drosophila* (Supplementary Fig. 9a, b, k–m). In conclusion, Ime4 and Mettl14 also counteracted the effect of BuGZ on the proliferation of ISC beyond YT521-B.

Different from YT521-B, BuGZ depletion in sorted *esg*+ cells did not affect the transcription level of Ime4 or Mettl14 mRNA (Supplementary Fig. 9n). However, we found that BuGZ interacted with Ime4 or Mettl14 protein, but not YT521-B (Supplementary Fig. 9o–q). To further explore the age-dependent interaction between Ime4/Mettl14 and BuGZ, we performed the immunoprecipitation assay by precipitating FLAG-BuGZ in midguts for endogenous Ime4 or Mettl14. Compared with young midgut, FLAG-BuGZ pulled down less Ime4 or Mettl14 in aged midgut when loading the same level of precipitated FLAG-BuGZ. Since BuGZ level increased, but Ime4/Mettl14 level showed no difference upon aging, our binding assay proposed that protein level enhancement of BuGZ might resulted in the reduced interaction between BuGZ and Ime4/Mettl14 in *Drosophila* midgut upon aging (supplementary Fig. 9r). To further demonstrate the mechanism of the interaction of Ime4/Mettl14 and BuGZ, we performed Co-Ip experiments in 293HEK cells. The results showed BuGZ$^{\Delta zinc}$ did not interact with Ime4 or Mettl14 (Supplementary Fig. 9s). Moreover, purified Ime4 or Mettl14 diminished the droplet formation of BuGZ but not BuGZ$^{\Delta zinc}$ in vitro (Supplementary Fig. 9t, u), suggesting that both *P8* and Ime4/Mettl14 bound to the zinc finger domain of BuGZ, and P8 and Ime4/Mettl14 might regulate BuGZ condensation with the same mechanism. Additionally, overexpressing either Ime4 or Mettl14 in ISCs suppressed the formation of BuGZ puncta without affecting its protein level in BuGZ$^{FL}$-overexpressing ISCs (Fig. 6a–d). Depletion of either Ime4 or Mettl14 in the ISCs enhanced BuGZ puncta formation in *Drosophila* overexpressing BuGZ$^{FL}$ (Fig. 6e–h). In addition, overexpression of either Ime4 or Mettl14 diminished the ISC hyperproliferation caused by BuGZ$^{FL}$ overexpression (Fig. 6i, j). Then, we further found depletion of either Ime4 or Mettl14 in *esg*+ cells reduced the YT521-B mRNA levels (Fig. 6k), whereas overexpressing either Ime4 or Mettl14 enhanced YT521-B mRNA levels in BuGZ$^{FL}$ overexpressing *esg*+ cells (Fig. 6l).

Therefore, we proposed that two m6A-related genes, Ime4 and Mettl14, inhibited BuGZ phase separation in *Drosophila* ISCs, to attenuate the transcription inhibitory property of BuGZ, promote the transcription of m6A reader YT521-B, the downstream of BuGZ, and prevent the proliferation of ISCs through the MAPK pathway (Fig. 6m).

## Discussion

Due to the impediments of observing the protein condensation processes in vivo and difficulties in verifying its physiological functions[2,4,7,9,10,43,44], the functions of protein phase separation in vivo, especially in tissue homeosltasis, organ regeneration, and individual aging, are largely unknown. Based on our investigation, the functions of BuGZ coacervation in regulating ISC activation and proliferation during aging and gut repair, imply that protein phase separation participates in tissue homeostasis, regeneration, and aging processes mediated by ASCs. Accordingly, we find that inhibiting BuGZ phase transition in ISCs of aged *Drosophila* improves intestinal functions and prolongs the lifespan of *Drosophila*, revealing that manipulating protein phase transition may delay or antagonize ASC-associated aging. Moreover, our findings reveal the physiological significance of the BuGZ condensate formation in DNA binding and transcriptional regulation in aging and gut repair and provide insights into the regulatory mechanisms underlying the BuGZ-mediated modulation of gene expression. Additionally, we report a mechanism that another *cis*-regulatory element (CRE) of genes the promotor facilitates the compartmentalization of its binding transcriptional condensate beyond enhancer and super-enhancer, other CREs including silencer and insulator may also possess feedback on regulating the coacervation behavior of their associated transcription factors.

Since RNA m6A modification behaves as the most common nucleotide modification of the eukaryotic mRNAs, previous studies have demonstrated that mRNA m6A modification widely affects various aspects of metabolism, homeostasis, ontogenesis, and pathema[45–49]. One previous study indicates that m6A reader YTHDF1 maintains stem cell traits during regeneration and tumorigenesis in intestines by mediating Wnt activation. In addition, since YTHDF1 upregulation is observed in both human and mouse intestinal cancer, YTHDF1 may behave as a biomarker for CRC[21]. Recently, another study uncovered that the m6A writer Mettl14, is required for modulating the homeostatic self-renewal in colonic stem cells. Mettl14 depletion in mouse colon results in colonic stem cell apoptosis, mucosal barrier dysfunction and severe colitis by modulating the NF-κB pathway[22]. In addition, previous reports indicated Mettl14-mediated GsdmC N6-adenomethylation is essential for the survival of Lgr5+ intestinal stem cells and the maintenance of normal colonic epithelial regeneration[23]. However, whether and how m6A regulators function in ISC-mediated gut regeneration upon injury and organ aging, remains largely unexplored. In our study, by systematically investigating the function of YT521-B in ISCs during aging in *Drosophila*, we find that m6A reader YT521-B plays an indispensable role in the ISC proliferation, gut homeostasis and lifespan. Under the transcriptional regulation of BuGZ, YT521-B functions as the upstream regulator of the EGFR/RAS/MAPK signaling pathway to mediate ISC proliferation. Moreover, BuGZ coacervation negatively regulates the mRNA transcription of YT521-B, whose promotor reversely enhances the phase separation of BuGZ. Meanwhile, we validate that m6A writers, Ime4 and Mettl14, inhibit the proliferation of ISC by suppressing the phase separation of BuGZ to regulate the transcription of YT521-B. Thus, our data reveals the relationship between m6A regulators and phase separation in controlling ISC proliferation during gut regeneration and aging. Meanwhile, we report a mechanism that controls the candidate TF coacervation by the downstream promotor and a way of probing the roles of m6A regulators in phase transition and regulation of ISCs. Additionally, our study not only deciphers a new regulatory mechanism of m6A reader YT521-B transcription but also a new function of m6A writer (Ime4, Mettl14) or reader (YT521-B) in ASCs homeostasis and aging.

Despite previous studies showing that m6A writer Mettl14 and reader YTHDF1 separately facilitate stem cell maintenance and regeneration in the m6A-dependent manner[21–23], some other studies propose that m6A regulators exert some physiological functions independent of their m6A modification activities. The m6A writer METTL3 plays a catalytic-independent role in recruiting eIF3 to the translation initiation complex to regulate the translation of oncogenic transcripts including EGFR and TAZ. This activity also enhances the PES1 mRNA translation in the cytoplasm of chronic myeloid leukemia cells and amplifies p53 signaling in response to cellular stress[50–55]. Another m6A writer, METTL16, also interacts with eIF3 to enhance mRNA translation during tumorigenesis in an m6A-independent manner[56]. Thus, future studies should evaluate whether the roles of m6A regulators (Ime4, Mettl14, and YT521-B) on ISC proliferation via the EGFR signaling pathways are mediated by the m6A modification. Future studies should also determine whether BuGZ and its phase separation act as the upstream regulator of m6A modification to affect m6A levels during gut regeneration and aging. Recently, the core m6A reader, YTHDF1/3, has been reported to promote stress granule formation through its phase transition[57,58]. Phase separation of m6A reader YTHDF1 promotes mRNA degradation by interacting with AGO2[59], while the m6A reader YTHDF2 undergoes m6A-mediated phase separation in mESCs[60]. Additionally, the phase separation of the m6A writer, METTL3, regulates the dynamic assembly of the mRNA m6A methyltransferase complex[61]. The predicted IDR domains within the m6A writers Ime4 (*Drosophila* ortholog of METTL3) and Mettl14 (data not shown) suggest that Ime4 and Mettl14 may adjust the ISC proliferation via their property of phase separation, and their

interferences in BuGZ coacervation also may rely on the interactions with BuGZ through their phase transition.

## Methods

### Cell lines

HEK293 cells (ATCC, #CRL-3216) were cultured at 37 °C, 5% CO₂ in Dulbecco's Modified Eagle Medium (DMEM, Gibco) supplemented with 10% fetal bovine serum (FBS, Gibco) and 5% antibiotics (penicillin/streptomycin, Invitrogen). *Spodoptera frugiperda* (Sf9, ATCC, #CRL-1711) cells used for the recombinant virus were grown in semi-suspension in Grace's Insect Medium (Supplemented) supplemented with 10% FBS at 27 °C. High Five™ (Hi5, from Sinobiological company) cells used for recombinant protein expression were grown in suspension in SIM HF Expression Medium at 27 °C, 150 rpm.

### *Drosophila* strains and fly genetics

The following *Drosophila* strains were used in this paper [obtained from the Bloomington *Drosophila* Stock Center (BDSC)]: *w^{1118}; PBac{BuGZ-GFP.FPTB}VK00033* (BDSC:67713), *386Y-GAL4* line: *w\*; P{w[+mW.hs] =GawB}386Y* (BDSC:25410), MARCM line: *w^{1118}; P{ry[+t7.2] =neoFRT}40 A/CyO;P{ry[+t7.2]=ey-FLP.N}6, ry[506]* (BDSC:8212), *BuGZ RNAi: y¹ v¹; P{TRiP.JF02830}attP2* (BDSC:27996), *UAS-Ime4* line: *y¹ w\*; P{UAS-Mettl3.HA.H}attP40/CyO* (BDSC:77882), *UAS-YT521-B* line: *y¹ w\*; P{UAS-Ythdc1.HA}attP40* (BDSC:77884), *UAS-EGFR-DN* line: *y¹ w\*; P{UAS-EGFR.DN.B}29-77-1; P{UAS-EGFR.DN.B}29-8-1* (BDSC:5364), *UAS-EGFR-CA* line: *w\*; P{Egfr.2.A887T.UAS}8-2* (BDSC:9533), *UAS-LacZ line*: *w˙; P{w[+mC]=UAS-lacZ.Exel}2* (BDSC:8529), *UAS-mCherry* line: *w\*; P{w[+mC]=UAS-mCherry.NLS}2; MKRS/TM6B, Tb¹* (BDSC:38425). *BuGZ RNAi: P{KK109387}VIE-260B* (VDRC: v104498), *Ime4 RNAi: w^{1118}; P{GD9882}v20969/TM3* (VDRC: v20969), *Mettl14 RNAi: w^{1118}; P{GD16300}v48560* (VDRC: v48560) were obtained from Vienna *Drosophila* Resource Center (VDRC). *YTS21-B RNAi:* THU1532, *Fl(2)d RNAi:* TH02871.N; *nito RNAi:* THU0697; *Vir RNAi:* TH02977.N; *CG6422 RNAi:* TH03062.N, MARCM line: *hsFLP, tub-GAL4, UAS-GFP-NLS/FM7; tub-Gal80, FRT40A/CyO* (TB00132) were obtained from Tsinghua Fly Center. *esg-GFP* (Carnegie Protein Trap line esg^{CB02017}), Flipout line: *yw, hsFLP; Act.FRT-CD2-FRT.GAL4, UAS-GFP^{S6ST}/CyO, UAS-Ras85D* line: *w^{1118};UAS-Ras85D.V12* were obtained from Allan Spradling. *esg-lacZ* (esg^{M5-4}) was obtained from Mark Van Doren. *NRE-lacZ, esg^{TS}-GAL4* line*: esg-GAL4, UAS-GFP, tub-Gal80^{TS}/CyO, ISC^{TS}-GAL4* line: *NRE-lacZ; esg-GAL4, UAS-GFP/CyO; tub-Gal80^{TS}, NRE-Gal80, NRE^{TS}-GAL4* line*: NRE-GAL4, UAS-GFP, tub-Gal80^{TS}/CyO* were obtained from Benjamin Ohlstein. Full Drosophila genotypes as they appear in each figure panel are listed in Supplementary Data 3.

### *Drosophila* husbandry

Unless otherwise noted, fly stocks were reared on the standard cornmeal/yeast medium (the recipe for 1 liter food contains: cornmeal 50 g, yeast 18.75 g, sucrose 80 g, glucose 20 g, agar 5 g, and propionic acid 30 ml). If not specifically mentioned, flies were maintained at 25 °C and 65% humidity on a 12 h light/dark cycle. Crosses not involving conditional expression of transgenes were incubated at room temperature. Crosses involving conditional Gal80ts-dependent expression of transgenes including RNAi or gene overexpression were maintained at 18 °C until adult offspring were enclosed. Adults were kept at 18 °C for 1–3 days before they were shifted to 29 °C to turn on the GAL4 system for another 7–10 days or 30 days before dissection. For the experiment, only mated females were used on the *Drosophila* midguts.

The *w^{1118}* strain was used as the control. BuGZ mutant line was generated with the CRISPR/Cas9 method using a synthetic guide RNA against the first common exon of all annotated BuGZ isoforms (sgRNA target sequence (5′–3′): GTCCGGTGTACAACTTCTTG), which was injected into *w^{1118}* embryos. After screening the offspring of the line containing sgRNA cross to the *vasa-cas9* flies which express Cas9 in the germline, 7 bp deleted (*BuGZ^8*) and 5 bp deleted (*BuGZ^3*) *BuGZ-null* flies

which result in a premature stop codon were generated by sequencing confirmed. We did not see a difference between *BuGZ^8/+* or *BuGZ^3/+* flies and *w^{1118}* flies, indicating that *BuGZ^8* and *BuGZ^3* are recessive mutations, and most time we used *BuGZ^8* flies for the following experiments.

Full indicated genotypes of flies used in this paper were listed in Supplementary Data 3.

### Generation of transgenic fly lines

Transgenic lines that we made house including: *UAS-dBuGZ^{FL}-HA, UAS-dBuGZ^{13S}-HA, UAS-dBuGZ^C-HA, UAS-dBuGZ^{CS}-HA, UAS-dBuGZ^{Δzinc}-HA, UAS-dMettl14-HA*. The final plasmids were cloned and verified by sequencing, then injected by UniHuaii Corporation (Zhuhai, China). Briefly, cDNA of dBuGZ^{FL} (isoform B from Flybase database), dBuGZ^{13S}, dBuGZ^{Δzinc}, dBuGZ^C, dBuGZ^{CS}, dMettl14 were cloned into the pEntry-3×HA vector and then subcloned into pTW vector (Gift from Allan Spradling). Primers used for these constructs are shown in Supplementary Data 1. Additionally, in order to observe the endogenous YT521-B transcription level, *YT521-B-luciferase* transgenic lines were made (Fangjing, China).

To get the knock-out line for BuGZ, the synthesized sgRNA sequence was cloned into a PMD18T vector to acquire U6 promoter. After the U6 promoter and sgRNA were amplified from the PMD18T vector by PCR, the products were subcloned together into the PCR8 vector with Golden Gate assembly. Then, the U6 promotor and sgRNA products were recombined into the attB vector by LR recombination reaction to generate the sgRNA construct. The final plasmids were injected by UniHuaii Corporation (Zhuhai, China).

### Clonal analyses

For MARCM clone analysis, we used the *BuGZ* mutant and FLP/FRT-mediated mitotic recombination technique. *FRT40A* (from Bloomington) was recombined with *BuGZ* mutant to produce *FRT40A, BuGZ^8-null* mutant, which was then crossed to *yw hsFLP, tub-Gal4, UAS-nls GFP/FM7; tubGal80 FRT40A/CyO* to obtain *hsFLP, tub-GAL4, UAS- nls GFP/+; tub-Gal80, FRT40A/ FRT40A, BuGZ^8* flies. Crosses were kept at 25 °C. To induce clones, adult flies, after eclosion 1-3 days old, were subjected to the 37 °C water bath for 1 h heat shock. Flies were maintained at 29 °C for 24 h within 2-3 days ACI after heat shock, and then these flies were dissected and observed at 10 days ACI.

For RNAi clone analysis, the FRT "Flipout (F/O)" method was used. Clones were generated by crossing *hsFLP; Act.FRT-CD2-FRT.GAL4, UAS-GFP/CyO* flies to the *BuGZ^{RNAi#1}, BuGZ^{RNAi#2}*, or *W^{1118}* flies. Crosses were maintained at 25 °C. Then, 1-3 days old adults were subjected to the 37 °C water bath for 1 h heat shock to induce clones. After heat shock, these flies were maintained at 29 °C for 24 h within 2-3 days ACI, and then these flies were dissected and observed at 10 days ACI.

### FACS, RT-qPCR, and western blotting analyses

As previously[62], the *Drosophila* adult midguts (R1-R5) were dissected into ice-cold PBS prepared with diethylpyrocarbonate (DEPC) water. After removing the foregut, malpighian tubules, hindgut and ovaries, the midguts were incubated with Trypsin-EDTA (diluting 100× 0.5% Trypsin-EDTA with DEPC-PBS) solution to each sample. Dissociated samples were pelleted, following resuspended in ice-cold DEPC-PBS and sorted using a FACS Aria II sorter (BD Biosciences) after filtering with 70 μm filters. Using *w^{1118}* midgut to set the fluorescence gate, GFP⁺ cells in the midgut of corresponding RNAi or indicated genotype flies driven by *esg^{TS}* were sorted out.

For each of the three biological replicates of RT-qPCR, about 100,000 *esg*-GFP⁺ cells were sorted. Total RNA was harvested using the RNA isolation kit (Cat.No.RE-03111, FOREGENE) based on the manufacturer's protocol. cDNA was synthesized using the PrimeScript RT reagent Kit (AG). Expression values were calculated using the 2^{−ΔΔCT} method and relative expression was normalized to *RpL15*. The

expression in the control sample was further normalized to 1. Primer sequences used for qPCR are available in Supplementary Data 1.

For each of the three biological replicates of western blotting, about 300,000 *esg*-GFP⁺ cells were sorted. Primary antibodies used for western blotting in this study include anti-BuGZ (rabbit, 1:1000), anti-α-tubulin (rabbit, 1:2000, Cell Signaling Technology, Cat# 2125), anti-YT521-B (rabbit, 1:1000). Secondary antibodies were horseradish peroxidase-conjugated goat anti-mouse (1:5000, Jackson ImmunoResearch Labs, Cat# 115-035-174), goat anti-rabbit (1:5000, Jackson ImmunoResearch Labs, Cat# 211-032-171).

## RNA-seq

RNA-sequencing was performed using *Drosophila* midguts overexpressing BuGZ in *esg*⁺ cells. The RNA sequencing was carried out by Shanghai OE Biotech Co., Ltd. (China). The sequence platform is based on novaseq 6000 (Illumina, San Diego, US) using a 150 bp paired-end run resulting in over 20 million reads per sample.

The initial quality control of the raw fastq files was performed using FastQC (v0.11.9, http://www.bioinformatics.babraham.ac.uk/projects/fastqc/), and the adapter sequences were removed by Trim Galore (v0.6.7, https://www.bioinformatics.babraham.ac.uk/projects/tri-m_galore/). The trimmed RNA-seq reads were aligned to the Drosophila reference sequence Ensembl build BDGP6 (https://support.illumina.com/sequencing/sequencing_software/igeno-me.html) with hisat2 (v2.20[63]). Aligned reads were then sorted by chromosome position using SAMtools (v1.15.1, https://sourceforge.net/projects/samtools/files/samtools/) and converted into bam files. Gene raw counts matrix per sample was computed using StringTie (v2.2.1, https://ccb.jhu.edu/software/stringtie/). Gene symbols were annotated by *Drosophila melanogaster* BDGP6 from Ensembl (https://support.illumina.com/sequencing/sequencing_soft-ware/igenome.html). Differentially expressed genes were determined using DESeq2 (v1.26.0[64],) with default parameters. Genes were considered differentially expressed if the absolute value of the log2 fold change was greater than 0.5 between modification (i.e., Overexpressed) and control at *p*adj < 0.05. Downstream pathway analysis was performed via clusterProfiler (v3.14.3[65],) using genes at *p*adj < 0.05.

## Immunofluorescence microscopy for *Drosophila* tissue

Midguts immunostaining was carried out as previous[62], female midguts were dissected in PBS and fixed in 4% EM-grade paraformaldehyde fixation buffer (100 mM glutamic acid, 25 mM KCl, 20 mM MgSO₄, 4 mM Na₂HPO₄, 1 mM MgCl₂, pH 7.4, 8% paraformaldehyde (Fisher Scientific, Cat# 5025997)) for 60 min at room temperature. For Delta immunostaining, dissected female guts are first fixed in fixation buffer (50% 4% paraformaldehyde plus 50% heptane) for 40 min and then fixed with methanol 10 min 3 times. For dpREK immunostaining, dissected female guts are first fixed in 8% paraformaldehyde buffer for 40 min and then fixed with methanol for 10 min at −20 °C. After 10 min washing (wash buffer: PBS plus 0.1% Triton X-100) 3 times, midguts were further blocked in 0.5% BSA (wash buffer plus 0.5% BSA) 30 min and then incubated with primary antibodies (anti-GFP, Abcam Cat# ab13970, 1:1000; anti-β-Galactosidase, Abcam Cat# ab9361,1:1000; anti-FLAG, Cell Signaling Technology Cat#14793, 1:1000; anti-phosphoHistone H3, Millipore Cat# 06-570, 1:1000; anti-Prospero, DSHB Cat# MR1A, 1:200; anti-Delta, DSHB Cat# C594, 1:100; anti-phospho-p44/42 MAPK (dpERK), Cell Signaling Technology Cat# 9101S, 1:1000; anti-Fibrillarin, ABclonal Cat# A13490, 1:200; anti-SUMO, ABclonal Cat# A2130, 1:1000; anti-γ-H2AvD, Rockland Cat# 600-401-914, 1:1000; anti-Hp1α, abcam Cat# ab77256, 1:1000; anti-H3-k27, Abclonal Cat# A2363; anti-Armadillo, DHSB Cat# N2 7A1, 1:100; anti-BuGZ, self-preparing, 1:1000.) diluted in the wash buffer overnight at 4 °C. The following day, after washing 10 min 3 times, the tissues were incubated with secondary antibodies (Alexa 488, Thermo Fisher Cat# A-11001, Cat# A-11034, Cat# A-11039; Alexa 568, Thermo Fisher

Cat# A-11004, Cat# A-11011, and Alexa 647, Thermo Fisher Cat# A-21235, Cat# A-21245, 1:2000) and DAPI (1 μg/ml) with for 2 h at room temperature. Three 10-min-washing steps were performed and midguts were mounted onto slides to observe using microscopy.

To carry out the temperature-dependent culture experiment in vitro, midguts were dissected in PBS at 25 °C. For 4 °C, the dissected guts were cultured in ice-cold Schneider's Drosophila Medium (Gibco) plus 5 μM MG132 at 4 °C for 1 h. For recovery from 4 to 25 °C, midguts incubated at 4 °C were shifted to room temperature for the next 1 h. After three different conditions treatments, midguts were fixed in fixation buffer and immunostained as mentioned above.

## Generation of antibodies

Antibodies against dBuGZ and dYT521-B were generated at ABclonal. For anti-dBuGZ sera, rabbit was immunized with a 122 amino-acid-long peptide (1–121 amino acids (AA)); for anti-YT521-B sera, rabbit was immunized with a 276 amino-acid-long peptide (29–304 amino acids (AA)). All serums were affinity-purified using peptide antigens cross-linked to sepharose columns.

## DNA in situ hybridization

For in situ hybridization, midguts were dissected in PBS and transferred to 4% formaldehyde for 10 min. After washing midguts 3 times in PBS, permeabilization was performed by using 0.5% Triton X-100 in PBS for 5 min at 4 °C, following 3 times PBS washes. Midguts were then incubated with preheated pre-hybridization buffer (RIBOBIO, C10910) at 37 °C for 30 min. After removing the pre-hybridization buffer, add probe hybridization mixture (100 μl of FISH Hybridization Buffer (RIBOBIO, C10910), 15 μl of FISH probe) to fully soak midguts, denatured at 78 °C for 5 min and transferred to 37 °C in the dark overnight. After removing the probe hybridization mixture, midguts were washed with washing buffer I (4×SSC, 0.1%Tween-20) 3 times, washing buffer II (2×SSC) 1 time, washing buffer III (1×SSC) 1 time, successively. After washing midguts 3 times in PBS, fixed with 4% PFA in PBS for another 10 min and washed 3 times in PBS, midguts were then incubated with the indicated primary antibody. Midguts were washed with PBS 3 times, followed by incubation with secondary antibody and DAPI for 2 h. Three 10-min washing steps were performed and midguts were mounted onto slides for observation. DNA FISH probes were custom-designed and generated by Wuhan servicebio technology CO to target the YT521-B promotor.

## Chip-qPCR

Chromatin was immunoprecipitated using a variation of the protocol described as the protocol of SimpleChIP® Plus Enzymatic Chromatin IP Kit (Magnetic Beads, cat #9005). In brief, approximately 50 corresponding genotype flies were grinded in liquid nitrogen and then were immediately fixed with 1% formaldehyde in Schneider's Drosophila Medium (Gibco). After filtering the sample, the cells were washed twice with ice-cold PBS, pelleted, and resuspended in Buffer A. After centrifuging, samples were resuspended in Buffer B. Cells were sonicated with a Ningbo Scientz Biotechnology Sonifier for 9 s/9 s 10 cycles at 30% power. After centrifuging at 13,000×*g* for 10 min, and the samples were diluted 10-fold in ChIP dilution buffer. The sample was incubated at 4 °C 4 h with antibodies against FLAG (1:50, Cat # 14793S)/ HA (1:50, Cat # 3724S) or Normal Rabbit IgG (1:50, Cat #2729). Complexes were precipitated with ChIP-Grade Protein G Magnetic Beads (Cat #9006). Beads were then washed sequentially with low-salt immune complex wash, high-salt immune complex wash (DNA wash buffer plus with NaCl). Immunoprecipitated chromatin was eluted in ChIP Elution Buffer, incubated at 65 °C for 30 min and then treated with proteinase K and 5 M NaCl for 6 h. At last, DNA was purified by Purification of DNA by centrifugation column as protocol. ChIP (enriched) and control (negative control) DNA samples were amplified by SimpleChIP® Universal qPCR Master Mix (Cat #88989).

## TUNEL assay

To detect apoptotic cells by TUNEL assay, female midguts were dissected in PBS and fixed with 4% paraformaldehyde in PBS for 24 h at 4 °C following washing with PBST (PBS with 0.1% Triton X-100) and immunostaining based on the in situ Cell Death Detection Kit manufacturer's protocol (Roche, Cat# 11684795910).

## EdU labeling

For EdU immunostaining, midguts were dissected in PBS and incubated in Schneider's Drosophila Medium with 50 μM EdU at room temperature. After 2 h, tissues were fixed as described above and stained for EdU using Cell-Light EdU Apollo567 In Vitro Kit (RIBOBIO, Cat# C10310-1) based on the manufacturer's protocol.

## Bromophenol blue treatment

Bromophenol blue assay was performed[30,66] and two categories of Drosophila midguts were observed[67] as previously described. After adding 250 μl of 2% Bromophenol blue sodium (pH indicator, Sigma, B5525) to a food vial which poked 4–6 holes using a pipet tip to allow full absorption, starved 2-h flies were put into the vial. After 24 h treatment, flies were immediately dissected to acquire images.

## Fly excretion measurement

For fly excretion measurement, as described previously[68], flies were drily starved for 2 h and put into Bromophenol blue vial food for 24 h, and then the deposits on the vial wall were imaged and quantified.

## Midgut measurement

Female Drosophila was orally administered BLM 24 h, after which guts were dissected in ice-cold 4% PBS. After dissection, the guts were mounted on slides. After being observed and imaged under fluorescence stereo microscopes (Leica M205 FA), Drosophila gut lengths were measured using LAS X software.

## Bleomycin and paraquat treatment

Bleomycin and paraquat treatment are carried out as previously[62]. Briefly, 25 μg/ml bleomycin (Aladdin, B107423) or 5 mM paraquat (Aladdin, M106761) dissolved in 5% (wt/vol) sucrose was added to chromatography paper that was cut into 3.7 × 5.8 cm strips. After being starved in empty vials for 2 h, flies then were transferred into vials with bleomycin-, or paraquat-solution-saturated chromatography paper. 5% (wt/vol) sucrose was used as controls (Mock). For microscopy, female guts were dissected to observe after 24 h treatment with paraquat or bleomycin feeding.

## Lifespan and survival experiments

For lifespan experiments, 80–100 mated female flies of the same genetics were separated and equally transferred into five vials containing the standard fly diet or standard fly diet supplement with 5 μg/ml bleomycin. Dead flies were counted and the food vial was exchanged every two or three days. The lifespan was repeated as three independent experiments.

## Luciferase activity assays

Firefly luciferase was measured using the Dual Luciferase Reporter Assay Kit (Vazyme, China, DL101-01). 300,000 esg$^+$ cells were sorted and frozen immediately with liquid nitrogen. Then, detecting the luciferase signals according to the manufacturer's protocol provided by the Dual Luciferase Reporter Assay Kit (Vazyme, Cat# DL101-01). Simultaneously, 20 μl of the lysate supernatant was collected for measurement of protein concentration by a BCA kit. The luciferase activity was determined by the absorbance normalized to protein concentration. Three replicates for each genotype and condition were performed.

## Cell transfection

For all transfections into HEK293T, Calcium Phosphate Cell Transfection Kit (Beyotime, Cat# C0508) was used. For transfections into Sf9 cells, Cellfectin® II Reagent (Gibco, Cat# 10362100) was used.

## Co-immunoprecipitation assay

For the co-immunoprecipitation assay in HEK293T, the plasmid of pCDNA3.1-FLAG-Ime4/Mettl14/YT521-B and pCDNA3.1-Myc-BuGZ$^{FL}$ or pCDNA3.1-FLAG-BuGZ$^{FL/}$ pCDNA3.1-FLAG-BuGZ$^{Δzinc}$ and pCDNA3.1-Myc-Ime4/Mettl14 were co-transfected into HEK293T cells seeded in a 10 cm cell culture dish. After 48-h transfection, HEK293 cells were harvested, lysed using lysis buffer (50 mM Tris-HCl, pH 7.5, 150 mM NaCl, 0.5% Triton X-100, with 1 mM PMSF) in a 1.5 ml centrifuge tube and rotated head-over-tail for 30 min at 4 °C. For immunoprecipitation assays in midguts, BuGZ-FLAG midguts were dissected in cold PBS. After homogenizing in liquid nitrogen, samples are lysed in 1% PBST buffer in a 1.5 ml centrifuge tube for 30 min on ice. The supernatant was collected after 10-min centrifugation at 1000×g at 4 °C. Protein concentrations were determined using a BCA kit. For immunoprecipitation, supernatant (Millipore, Cat# A2220) was added into anti-FLAG M2 gel and rotated head-over-tail overnight at 4 °C for 4 h. The beads were washed 3 times for 10 min with lysis buffer. Immunoprecipitated proteins and input samples were analyzed by western blotting after elution with SDS-PAGE sample buffer.

## Protein expression and purification

pet28a$^+$-dBuGZ$^{FL}$/dBuGZ$^{Δzinc}$/GFP construct was transformed into BL21 (DE3). To induce protein expression, 1 mM IPTG was added to the culture upon OD600 reached 0.6–1, then cooled the culture to 16 °C with shaking for 16 h. After centrifugation at 3800 rpm for 20 min at 4 °C, bacteria were collected. The bacteria were broken by sonicating (SCIENTZ-IID) until the lysis buffer (50 mM NaH$_2$PO$_4$, 300 mM NaCl, 10 mM imidazole, 1 mM PMSF, cocktail, 1:100, pH 7.8) became clear (-25 min), and the lysate was centrifuged at 12000 rpm for 30 min at 4 °C. The supernatant was transferred to a column (Glass Econo-Column®, Bio-RAD) containing 2 ml of 50% slurry of Ni-NTA beads (Qiagen). After rotating the mixture at 4 °C for 4 h, the column was washed with 100 ml wash buffer 1 (50 mM NaH$_2$PO$_4$, 300 mM NaCl, 20 mM imidazole, pH 7.8) and following 100 ml wash buffer 2 with 25 mM imidazole. At last, proteins were eluted with 16 ml Elution buffer (50 mM NaH$_2$PO$_4$, 300 mM NaCl, 250 mM imidazole, pH 7.8) in 1 ml fractions. The elution was concentrated to 100-350 μM with Amicon Ultra 30 K device (Millipore) at 4 °C and exchanged with PBS buffer at the same time. The protein stocks were divided into 10 μl aliquots, snap-frozen in liquid nitrogen, and stored at −80 °C.

For expression and purification of His-GFP-3×GGGS-dIme4/dMettl14, the manual of Bac-to-Bac Baculovirus Expression system (Invitrogen) was used. After getting recombined P1-P3 virus from Sf9, and 600 ml of High Five cells suspension was used to express the corresponding protein. High Five cells were harvested and lysed in 40 ml of binding buffer (20 mM phosphate buffer, pH 7.4, 1 mM MgCl2, 1 mM β-mercaptoethanol, 1% Triton X-100, 0.5 M NaCl, 25 mM imidazole, 10% glycerol) supplemented with 1 mM phenylmethylsulfonyl fluoride (PMSF). The cell lysate was centrifuged at 12,000 rpm for 30 min at 4 °C. The supernatant was loaded onto a column packed with 4 ml of 50% slurry of Ni-NTA beads (Qiagen) and followed by washing with 20 volumes of His-binding buffer. The protein was then eluted with 8×2 ml of His-elution buffer (same as His-binding buffer except with 500 mM imidazole). The eluted fractions were combined and concentrated by Amicon Ultra 30 K device (Millipore) to a final volume of 2–3 ml. Samples were exchanged with PBS buffer twice at 4 °C, divided into 10 μl aliquots, snap-frozen in liquid nitrogen, and stored at −80 °C.

All protein purification steps were performed at 4 °C. The purified protein was quantified using an ND-2000C NanoDrop spectrophotometer (NanoDrop Technologies) with OD 280.

### In vitro phase separation assays

For the standard phase-separation assay, 5 µl of His-dBuGZ[FL], His-dBuGZ[Δzinc] proteins at the indicated concentrations were prepared by diluting the stock solutions into ice-cold PBS buffer. After incubation of the mixtures at 37 °C for 5 min, the droplets were immediately observed.

Temperature-dependent phase separation experiments were performed by incubating the indicated protein mixture at 37, 30, 25 and 4 °C for 5 min after removal from ice.

To visualize the salt hydronium concentration-dependent phase transitions, His-dBuGZ dissolved in storage buffer were mixed at a desired ratio and concentration, and the concentration of NaCl was adjusted with buffer containing 25 mM HEPES pH 7.5, 1 M NaCl and 1 mM DTT.

To assay whether His-GFP-dIme4, His-GFP-dMettl14 regulate His-dBuGZ droplets, we mixed 30 µM His-dBuGZ-FL/Δzinc with 30 µM His-GFP, His-GFP-dIme4, His-GFP-dMettl14 on ice.

DNA-dependent droplet formation experiments were performed by adding 10 µM indicated DNA to 16.7 µM or 8.3 µM His-dBuGZ[FL] or 16.7 µM His-dBuGZ[Δzinc] in the storage buffer. The solution was incubated at 37 °C for 5 min and immediately the state of the droplet was observed.

Formation and fusion of liquid droplets was observed by bright field of fluorescence microscopy using a 40× objective in Leica DM6-B microscope.

Fluorescence recovery after photobleaching (FRAP) was performed on scarlet-BuGZ droplets. Briefly, one BuGZ droplet was bleached and the fluorescence recovery was monitored. 10 µM P8 was added to 20 µM His-scarlet-dBuGZ[FL].

### Percentage of cells, fluorescence intensity and area of puncta statistics

The results of the immunofluorescence image were analyzed based on z stacks acquired with confocal microscopy.

The percentage of indicated genotype cells is measured by calculating all the positive cells in a picture and then normalized by DAPI counts.

Fluorescence intensity of the region of interest (ROI) or cell was calculated using LAS X software. The methods involved are as below: Integrated Density = Integrated Density of ROI or cell−Integrated Density of background region/Area of background region × Area of ROI or cell.

The area of puncta was calculated using Image J.

### Statistics and reproducibility

All experiments' statistical analyses were performed using Prism 7 (GraphPad Software). Statistical significance was shown are means ± SD of three independent replicates. Differences between means were assessed with unpaired two-tailed Student's t-tests.

### Reporting summary

Further information on research design is available in the Nature Portfolio Reporting Summary linked to this article.

## Data availability

The RNA-seq data generated in this study have been deposited in the Sequence ReadArchive repository database under accession code PRJNA885069. The up- and down-regulated genes generated in this study are provided in the Supplementary datasheet (Supplementary Data 2). Source data are provided with this paper.

## Code availability

Custom codes are available from the corresponding author upon request.

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

## Acknowledgements

We thank BDSC, VDRC, Tsinghua Fly Center for fly strains, and DSHB for antibodies. This work was supported by: National Natural Science Foundation of China (32090043, (H.J.)); National Key Basic Research Program of China (2020YFA0803602 (H.C.)); National Key Basic Research Program of China (2018YFA0108302 (H.J.) and 2018YFA0108301 (H.C.)); National Natural Science Foundation of China (31622031 (H.C.), 31671254 (H.C.), 91749110 (H.C.) and 31771505 (H.J.));

National Clinical Research Center for Geriatrics, West China Hospital, Sichuan University (Z20201009 (H.J.), Z20191011 (H.J.), Z20201006 (H.C.) and Z2023YY003 (H.J.)); 1.3.5 project for disciplines of excellence, West China Hospital, Sichuan University (ZYYC20001 (H.J.), ZYYC20024 (H.C.), ZYGD20010 (H.J.)); Sichuan Science and Technology Program, the Central Government Guides Local Science and Technology Development Projects, China (Grant No. 2022ZYD0078 (H.J.)); Sichuan Science and Technology Program (Grant No. 2023YFQ0008 (H.J.)); Project of Max Cynader Academy of Brain Workstation, WCHSCU (HXYS19005 (H.J.)); Science and technology department of Tibet, the central government guides the local science and technology development fund project (XZ202102YD0026C (H.J.)).

## Author contributions

Conceptualization: H.J. and H.C. Methodology: H.J., H.C., Q.Z., M.L., S.Y., K.D., T.F. and M.J. Investigation: Q.Z., K.D., M.L. and W.X. Visualization: Q.Z., M.L., S.Y., W.X, X.S., and Z.L. Pattern drawing: Q.Z., Funding acquisition: H.C. and H.J. Project administration: H.C. and H.J. Supervision: H.C. and H.J. Writing—original draft: Q.Z., H.C. and H.J. Writing—review & editing: Q.Z., K.D., H.C., and H.J.

## Competing interests

The authors declare no competing interests.
