## [Peer Review File · Nature Communications]

Phase separation of BuGZ regulates gut regeneration and aging through interaction with m6A regulatorsREVIEWER COMMENTS

Reviewer #1 (Remarks to the Author):

Ms. Nature Communications

Title: Phase separation regulates gut regeneration and aging by m6A regulators

General Comments

This manuscript reports that the coacervating mitotic effector BuGZ shows phase condensation in the nuclei of aged and injured *Drosophila* intestinal stem cells, and this property of BuGZ promotes the stem cell proliferation that influences gut repair and the life span of *Drosophila*. This work further reports that YT521-B, a m6A reader protein, functions as a downstream of BuGZ to induce the MAPK pathway to promote ISC proliferation, and a YT521-B promoter fragment promotes BuGZ phase separation. Moreover, it further shows other m6A players – m6A writers Ime4 and Mettl14 proteins – are also involved in BuGZ phase separation and thus in the regulation of ISC-dependent gut regeneration. There are no clear reasons and logics why these m6A players are not involved in conventional m6A regulation rather all involved in the regulation of this particular protein's phase condensation by totally different mechanisms – participation of a promoter DNA fragment in the case of YT521-B to promote BuGZ condensation and direct protein participation in the case of Ime4 and Mettl14 to suppress condensation.

Specific comments

1. There are a large number of genetic *Drosophila* strains (knockout or overexpression) or RNAi knockdown strains used in the study, but none of them has validation data for the genetic or expression changes.
2. The figures are hard to follow. For example, in figure 1, what does DI represent? What GFP does represent? There are no explanations in the text or in the legend. Similar problems are seen in other figures. The authors are suggested to go over all figures to make sure all labels are understandable to readers.
3. Fig. 1J, controls without MG132 treatment are not seen. In Fig. 1a, BuGZ shows condensation on 30D not on 10D, but in Fig. 1j, BuGZ condensation already occurs on 7D. The same is seen in Figure 3 on 7D. Why the difference?
4. Are MARCM (in the text and legends) and MACAM (in the Figure) the same thing?

5. The authors states that the reason why this work focuses on m6A players is that they have shown a strong relationship between BuGZ and RNA-associated proteins in Ref. 31, but there is nothing indicating such a finding in Ref. 31. So there appears to be a pure guess to link BuGZ to YT521-B and later to Ime4 and Mettl14. If so, why not guess other RNA-binding proteins or other proteins? There is not a clear logic for this guessing.
6. In situ hybridization using p8 DNA fragment only showed colocalization with BuGZ (Fig. 5). In theory, p8 DNA probe should bind to all cell genomes that have YT521-B promoter. The specificity is questionable.
7. If BuGZ condensation increases with ageing or injury, and a YT521-B promoter fragment binds to BuGZ to increases its condensation (leading to suppression of YT521-B expression), then can one assume BuGZ and p8 interaction increases with ageing? Can the author demonstrate that?
8. How does YT521-B affect the MAPK pathway?
9. Does Ime4 or Mettl14 expression change with ageing or injury? What controls age-dependent Ime4 or Mettl14 and BuGZ interaction? If Ime4 or Mettl14 have m6A-independent activity in this case, does m6A status in ISCs change with ageing or injury?
10. Do DNA (p8) and protein (Ime4 or Mettl14) regulate BuGZ condensation in different mechanisms?
11. The discussion is short. There have been quite a few recent publications that demonstrate critical roles of m6A regulators in the regulation of mouse gut homeostasis in the classic m6A regulation manner. At least these findings should be discussed with regards to their differences from this work. More discussion about the mechanism and implication is suggested, particularly in the context of gut homeostasis, ageing and injury.

Reviewer #2 (Remarks to the Author):

In this manuscript, Zhang et al. report that BuGZ regulates injury induced ISC proliferation through YT521-B and the MAPK pathway. In addition, the authors demonstrate that LLPS of BuGZ is regulated by YT521-B promoter and Ime4 and Mettl14. Overall, the findings in this study are interesting. However, there are some concerns that should be addressed.

1. On page 5, the authors state that "BuGZ enhanced ISC proliferation to repair the damaged gut epithelial after injury but was not required for ISC basic maintenance." However, it is unclear how the authors draw the conclusion that BuGZ enhanced ISC proliferation. They did not show that overexpression of BuGZ induced ISC proliferation. Furthermore, the authors' claim that BuGZ is not required for ISC basic maintenance is not well-supported. They did not examine stem cell fate markers in their experiments. Therefore, it is better to rephrase their conclusion and examine the stem cell fate marker.

2. The authors concluded that BuGZ mediates ISC proliferation and gut repair after injury by performing cell-autonomous functions. However, the authors did not provide clear evidence to support this claim. To strengthen their argument, they could demonstrate that RNAi knockdown of BuGZ in enteroendocrine cells or enteroblasts completely blocks BuGZ expression in ISCs. This would help to confirm that BuGZ functions in a cell-autonomous manner to regulate ISC proliferation and gut repair after injury.

3. In Supplementary Fig. 3j, is the NRE-LacZ expressed in polyploid cells in the control?

4. The authors state that "increased expression of BuGZ led to ISC hyperproliferation." However, the data presented in Figure 2N show that the number of cells positive for the proliferation marker PH3 is less than 35 in BuGZ-overexpressing ISCs. Therefore, the term "hyperproliferation" may not be appropriate. Instead, the authors could state that BuGZ overexpression leads to an increase in ISC proliferation.

5. The relationship between BuGZ and YT521-B is not clear in the manuscript. To strengthen the authors' claim that BuGZ regulates YT521-B expression, the authors could perform a chromatin immunoprecipitation (ChIP) experiment to examine whether BuGZ can bind to the YT521-B promoter region. This would provide direct evidence of the interaction between BuGZ and YT521-B.

6. The mechanism by which BuGZ and YT521-B activate the MAPK pathway is critical for explaining the function of BuGZ and YT521-B in regulating ISC proliferation and gut repair after injury. The authors should provide more details on how BuGZ and YT521-B activate the MAPK pathway. This information would help to elucidate the downstream signaling pathways that are regulated by BuGZ and YT521-B and provide a more complete understanding of their roles in ISC proliferation and gut repair after injury.

Minor points:

1. "BuGZ13S overexpression in *esg*⁺ cells showed fewer effects on the ISC proliferation than BuGZFL (Fig. 3b, f, g)" in P8. It should be rephrased.

2. For "contrary to the BuGZ RNAi phenotype that we demonstrated in the asymmetric cell division of ISC after injury" in P6, the sentence is confusing.

Reviewer #3 (Remarks to the Author):

The manuscript entitled “Phase separation regulates gut regeneration and aging by m6A regulators” authored by Zhang Q. et al. describes the molecular mechanism of how liquid-liquid phase separation plays a role in the regeneration and aging of gut in fly. They focused on BuGZ protein, which had been known to form a liquid-like condensate in mitotic cells, and examined its role in intestinal stem cells. They performed a number of experiments and found that BuGZ forms a condensate in interphase intestinal stem cell nucleus and affects fly gut repair and longevity. They also identified down-stream factors such as YT512-B and demonstrated that BuGZ play a role in the transcriptional regulation of YT521-B and *Ime4/Mett14* genes. From mechanistic point of view, the manuscript presents a new pathway and mechanism of BuGZ-dependent gut regeneration. On the other hand, in terms of a role of liquid-liquid phase separation in this process, most of the data are still descriptive and do not provide new mechanism or concept of phase separation in gut repair. If the authors are willing to show a critical role of BuGZ “phase behavior” in gut repair (this is what they emphasize in Introduction), they have to clarify the following issues.

1. Because BuGZ had been identified as a microtubule-regulating protein, it was surprising that BuGZ was involved in the transcriptional regulation. As shown in Fig. 1a and 1j (and many others), BuGZ formed several huge condensates in ISC nuclei. It seems that the size, shape and number of foci in a nucleus varied a lot from cells to cells. If BuGZ is involved in the transcriptional regulation, how a large variation of the size and number of foci can be explained? It is more reasonable to expect a number of target genes not only YT521-B should exist. If the authors want to understand the role of BuGZ condensate in the transcriptional regulation, they have to clarify this. I wonder if they ever performed chromatin immunoprecipitation to identify all possible target genes.
2. The authors proposed that BuGZ binds to the promoter region of YT521-B gene, assemble there, and finally form a large condensate. Then, I wonder what is the physiological significance of the formation of such a large condensate, rather than a simple “binding”. Generally, transcriptional control can be achieved by simple binding of a single or a few copies of a transcription factor to the promoter. If the authors want to understand the role of BuGZ phase separation, they have to show the evidence for physiological role of condensate formation, which cannot be achieved by a simple binding.
3. The morphological variation of BuGZ condensates in a cell reminds various nuclear speckles rather than transcription condensates, which are generally much smaller. The authors examined only PML bodies with anti-SUMO antibody. However, I wonder whether BuGZ condensate colocalize with other known nuclear speckles. Because BuGZ has been demonstrated to be involved in mRNA splicing (as they mentioned in Introduction), this may be possible. Although I understand that the BuGZ condensate is huge and not similar to any known nuclear speckles, these possibilities should be tested.

Reply to Reviewer #1

General Comments:

This manuscript reports that the coacervating mitotic effector BuGZ shows phase condensation in the nuclei of aged and injured *Drosophila* intestinal stem cells, and this property of BuGZ promotes the stem cell proliferation that influences gut repair and the life span of *Drosophila*. This work further reports that YT521-B, a m⁶A reader protein, functions as a downstream of BuGZ to induce the MAPK pathway to promote ISC proliferation, and a YT521-B promoter fragment promotes BuGZ phase separation. Moreover, it further shows other m⁶A players – m⁶A writers Ime4 and Mettl14 proteins – are also involved in BuGZ phase separation and thus in the regulation of ISC-dependent gut regeneration. There are no clear reasons and logics why these m⁶A players are not involved in conventional m⁶A regulation rather all involved in the regulation of this particular protein's phase condensation by totally different mechanisms – participation of a promoter DNA fragment in the case of YT521-B to promote BuGZ condensation and direct protein participation in the case of Ime4 and Mettl14 to suppress condensation.

Response:

*We appreciate for your clear and detailed feedback. We apologize that we did not clearly state the reasons and logic why these three m⁶A players are included in this study in our original manuscript. In fact, our previous studies^{1, 2} identified BuGZ binding proteins in ESCs with mass spectrometry, and showed strong interactions of BuGZ with RNA-associated proteins, especially with m⁶A-related proteins (Hnrpa2b1, Hnmpa1, Igf2bp3) highlighted with yellow color in the table (Supplementary Fig. 5a in the revised manuscript), inferring that m⁶A-related proteins might be involved in the physiological functions of BuGZ in stem cells. Then, to investigate the mechanism by which BuGZ regulated ISC proliferation in *Drosophila*, we performed a candidate genetic screen with these m⁶A-associated genes to uncover the potential genes that could rescue BuGZ RNAi phenotype in regulating the ISC proliferation after gut injury, which revealed that only Ime4, Mettl14, and YT521-B depletions reversed the BuGZ*

depletion phenotype in the screen (Supplementary Fig. 5b in the revised manuscript). Thus, we next chose these three candidates to unravel the mechanism by which BuGZ controlled ISC proliferation during gut regeneration and aging. Meanwhile, we believe the revisions we made as detailed below have addressed the other concerns raised by this reviewer.

Specific comments

1. There are a large number of genetic Drosophila strains (knockout or overexpression) or RNAi knockdown strains used in the study, but none of them has validation data for the genetic or expression changes.

Response:

We really appreciate this suggestion. In the revised manuscript, we have included validation data for the genetic or expression changes (see supplementary Fig.3a, b, i; supplementary Fig.5c; supplementary Fig.9a, b). These data indicated that all genetic Drosophila strains (knockout, RNAi, and overexpression) were effective and relevant results are shown as follows.

BuGZ RNAi, overexpression, knockout validation (supplementary Fig.3a, b, i)

Validation of YT521-B RNAi and overexpression (supplementary Fig.5c)

Validation of Ime4/Mett14 RNAi and overexpression (supplementary Fig.9a, b)

2. The figures are hard to follow. For example, in figure 1, what does DI represent? What GFP does represent? There are no explanations in the text or in the legend. Similar problems are seen in other figures. The authors are suggested to go over all figures to make sure all labels are understandable to readers.

Response:

We thank this reviewer for this critique and advice. We apologize for not describing these labels clearly in our original manuscript. The full name of “DI” is “Delta”, which acts as the Notch signaling ligand and is typically used for labeling ISCs in Drosophila³. The “GFP” represents “esg-GFP” which marks the intestinal stem and progenitor cells⁴. In the revised manuscript, we have carefully and thoroughly proofread the manuscript in figures, text, and figure legend to make sure all labels are understandable.

3. Fig. 1J, controls without MG132 treatment are not seen. In Fig. 1a, BuGZ shows condensation on 30D not on 10D, but in Fig. 1j, BuGZ condensation already occurs on 7D. The same is seen in Figure 3 on 7D. Why the difference?

Response:

Thanks for this question. In the revised manuscript, we have included the data for control without MG132 treatment (supplementary Fig. 1f-i). Immunofluorescence

intensity analyses showed that BuGZ protein level decreased at low temperature without MG132 treatment, whereas BuGZ puncta still diffused when the midguts were transferred from 25 °C to 4 °C. Moreover, the diffused BuGZ protein concentrated again to form more but smaller puncta when these low temperature-cultured midguts were transferred back to 25 °C (supplementary Fig. 1f and 1h-i).

The change of BuGZ puncta at low temperature without adding MG132 (supplementary Fig.1f-i)

In this study, endogenous BuGZ formed nuclear puncta from 30D but not 10D upon aging (Fig.1a). However, overexpressing BuGZ in *esg*⁺ cells resulted in concentration-dependent BuGZ condensation in younger flies (7D), using the *esg*-Gal4, UAS-GFP, *tub*-Gal80^{TS} system⁵ (Fig.1j).

4. Are MARCM (in the text and legends) and MACAM (in the Figure) the same thing?

Response:

We feel quite sorry about the typos. We have corrected the MACAM with MARCM in our revised manuscript including the figures, text, and figure legend.

5. The authors states that the reason why this work focuses on m⁶A players is that they have shown a strong relationship between BuGZ and RNA-associated proteins in Ref. 31, but there is nothing indicating such a finding in Ref. 31. So there appears to be a pure guess to link BuGZ to YT521-B and later to Ime4 and Mettl14. If so, why not guess other RNA-binding proteins or other proteins? There is not a clear logic for this guessing.

Response:

Thanks for this question. In Ref.31, our previous study showed a mass spectrometry detection for BuGZ binding proteins in ESCs in the Table S1. We apologize that we did not clearly state the reasons and logic why these three m⁶A players are included in this study in our original manuscript. In fact, our previous studies^{1, 2} identified BuGZ binding proteins in ESCs with mass spectrometry, and showed strong interactions of BuGZ with RNA-associated proteins, especially with m⁶A-related proteins (Hnrpa2b1, Hnmpa1, Igf2bp3) highlighted with yellow color in the table (Supplementary Fig. 5a in the revised manuscript), inferring that m⁶A-related proteins might be involved in the physiological functions of BuGZ in stem cells. Then, to investigate the mechanism by which BuGZ regulated ISC proliferation in Drosophila, we performed a candidate genetic screen with these m⁶A-associated genes to uncover the potential genes that could rescue BuGZ RNAi phenotype in regulating the ISC proliferation after gut injury, which revealed that only Ime4, Mettl14, and YT521-B depletions reversed the BuGZ depletion phenotype in the screen (Supplementary Fig. 5b in the revised manuscript). Thus, we next chose these three candidates to unravel the mechanism by which BuGZ controlled ISC proliferation during gut regeneration and aging.

a		b		
Gene name		esg ^{TS}		
Nono		RNAi	Phenotype (-BLM)	Rescued the phenotype of BuGZ RNAi (+BLM)
Skiv212		lme4	Promote proliferation of ISCs	Y(***)
Sfpq		Mettl14	Promote proliferation of ISCs	Y(***)
Ddx5		Fl(2)d	No phenotype	N
Pabp1		Nito	No phenotype	N
Ybx1		Vir	No phenotype	N
Hnrpa2b1;Hnrpa2b1		YT521-B	Promote proliferation of ISCs	Y(***)
Rbm14		CG6422	No phenotype	N
Hnrpa1		RNAi is driven by esg ^{TS} . Y means the phenotype of BuGZ RNAi is rescued by indicated gene. N means the phenotype of BuGZ RNAi is not rescued by indicated gene. Student's t-test, *p<0.05, **p<0.01, ***p<0.001.		
Hnrmpu				
Hnrnpa3				
Igf2bp3				
Sap155				
Sf3b3				
Sf3a1				
Hnrpq				
Sfrs1				
Sfrs6				

m⁶A-related protein in mass spectrometry and screen results (supplementary Fig.5a-b)

6. In situ hybridization using p8 DNA fragment only showed colocalization with BuGZ (Fig. 5). In theory, p8 DNA probe should bind to all cell genomes that have YT521-B promoter. The specificity is questionable.

Response:

We thank this reviewer for this valuable comment. In the result of Fig. 5b, despite the P8 DNA probe should bind to various cell genomes, the BuGZ remarkable puncta induced by BuGZ overexpression, might recruit more P8 DNA probes when compared with endogenous BuGZ nuclear condensate, resulting in more obvious co-localization. Moreover, to present results clearly, we adjusted the contrast of the images during analyzing the co-localization.

7. If BuGZ condensation increases with ageing or injury, and a YT521-B promoter fragment binds to BuGZ to increases its condensation (leading to suppression of YT521-B expression), then can one assume BuGZ and p8 interaction increases with ageing? Can the author demonstrate that?

Response:

We are grateful for this great suggestion on improving the integrity of our study. To address this question, we analyzed the interaction between BuGZ and P⁸ fragment by Chip-qPCR in the young or old guts of transgenic BuGZ-Flag fly (Fig. 5a). The results demonstrated that the interaction between BuGZ and P⁸ was enhanced upon aging.

Results of Chip-qPCR with immunoprecipitating BuGZ in young and old flies (Fig. 5a)

8. How does YT521-B affect the MAPK pathway?

Response:

We thank the reviewer for this constructive suggestion. Since YT521-B did not act as the typical transcriptional factor/suppressor, there was a deficiency of how YT521-B affects the MAPK pathway in our original manuscript. To study the detailed mechanism of the YT521-B modulation in the MAPK pathway, we examined the ERK activity assessed by immunostaining against the double phosphorylated ERK kinase (dpERK), which behaved as the key downstream target of the MAPK pathway⁶. Our data showed YT521-B overexpression inhibited the ERK activation caused by BuGZ overexpression (supplementary Fig. 7h).

YT521-B regulates the MAPK pathway by ERK signal (supplementary Fig. 7h)

9. Does *Ime4* or *Mettl14* expression change with ageing or injury? What controls age-dependent *Ime4* or *Mettl14* and BuGZ interaction? If *Ime4* or *Mettl14* have m6A-independent activity in this case, does m6A status in ISCs change with ageing or injury?

Response:

*We are grateful for these questions. Accordingly, we found that both transcription and protein levels of *Ime4/Mettl14* were invariant during aging (supplementary Fig. 9r, figure A, B), whereas the mRNA level of *Ime4/Mettl14* was elevated upon gut injury (figure A, B).*

*The level of *Ime4/Mettl14* in aging or gut injury (figure A, B).*

To further explore the mechanism of age-dependent interaction between *Ime4/Mettl14* and *BuGZ*, we performed the immunoprecipitation assay by precipitating *FLAG-BuGZ* in midguts for endogenous *Ime4* or *Mettl14*. Compared with young midgut, *FLAG-BuGZ* pulled down less *Ime4* or *Mettl14* in aged midgut when loading the same level of precipitated *FLAG-BuGZ* (supplementary Fig. 9r). Since *BuGZ* level increased, but *Ime4/Mettl14* level showed no difference upon aging, our binding assay proposed that protein level enhancement of *BuGZ* might result in the reduced interaction between *BuGZ* and *Ime4/Mettl14* in *Drosophila* midgut upon aging.

Ime4/Mettl14 interacted with *BuGZ* upon gut repair (supplementary Fig. 9r)

Additionally, we investigated the m^6A status upon aging or injury, which revealed that the m^6A level increased upon both aging and gut injury (figure C). Based on all these data, the levels of *Ime4*, *Mettl14* and m^6A all increased during the process of injury, implying that *Ime4/Mettl14* might regulate the ISC proliferation in the m^6A -dependent manner. During the aging process, despite the levels of *Ime4* and *Mettl14* did not change, their m^6A modification activities might be also involved in the ISC modulation since the total m^6A level was enhanced upon aging. Furthermore, the process of aging usually took a long time when compared with gut injury, which might explain why *Ime4/Mettl14* might be required for ISC proliferation via the m^6A modification with their invariant levels.

The m⁶A level upon aging or gut injury (figure C)

10. Do DNA (p8) and protein (Ime4 or Mettl14) regulate BuGZ condensation in different mechanisms?

Response:

Thanks for this question. In our original manuscript, we showed that P⁸ fragment enhanced BuGZ phase separation by binding to its zinc finger domain (Fig. 5b, f and supplementary Fig. 8g-h). However, the domain of BuGZ, with which Ime4 and Mettl14 interacted to inhibit BuGZ condensate, remained unclear and needed to be further explored. Thus, we performed Co-IP experiments in 293HEK cells., the results showed BuGZ-Δzinc did not interact with Ime4 or Mettl14 (supplementary Fig. 9s). Meanwhile, different from wild-type BuGZ, the phase separation of BuGZ-Δzinc was not affected by Ime4 or Mettl14 (supplementary Fig. 9u). All these data suggested that both P⁸ and Ime4/Mettl14 bound to the zinc finger domain of BuGZ, and P⁸ and Ime4/Mettl14 might regulate BuGZ condensation with the same mechanism.

Ime4/Mettl14 interacts with BuGZ-FL instead of BuGZ-Δzinc (supplementary

Fig. 9s)

The effect of Ime4/Mettl14 protein on phase separation of BuGZ or BuGZ-Δzinc

(supplementary Fig. 9u)

11. The discussion is short. There have been quite a few recent publications that demonstrate critical roles of m⁶A regulators in the regulation of mouse gut homeostasis in the classic m⁶A regulation manner. At least these findings should be discussed with regards to their differences from this work. More discussion about the mechanism and implication is suggested, particularly in the context of gut homeostasis, ageing and injury.

Response:

We really appreciate this great suggestion. According to this reviewer's comments, we have added more discussions in our revised manuscript including the prominent roles of m⁶A regulators in the regulation of mouse gut homeostasis and the mechanism and significance of BuGZ in regulating ISC proliferation in the context of gut homeostasis and aging.

Reply to Reviewer #2

In this manuscript, Zhang et al. report that BuGZ regulates injury induced ISC proliferation through YT521-B and the MAPK pathway. In addition, the authors demonstrate that LLPS of BuGZ is regulated by YT521-B promoter and Ime4 and Mett114. Overall, the findings in this study are interesting. However, there are some concerns that should be addressed.

Response:

We are grateful for these well-encouraged and positive comments. We believe the revisions we made as detailed below have addressed all the concerns raised and further strengthened our conclusions.

1. On page 5, the authors state that "BuGZ enhanced ISC proliferation to repair the damaged gut epithelial after injury but was not required for ISC basic maintenance." However, it is unclear how the authors draw the conclusion that BuGZ enhanced ISC proliferation. They did not show that overexpression of BuGZ induced ISC proliferation. Furthermore, the authors' claim that BuGZ is not required for ISC basic maintenance is not well-supported. They did not examine stem cell fate markers in their experiments. Therefore, it is better to rephrase their conclusion and examine the stem cell fate marker.

Response:

*We thank the reviewer for these suggestions and apologize for not clarifying it clearly. In the original manuscript, we indicated BuGZ RNAi in ISCs inhibits the proliferation ISCs upon gut repair (Fig.2a-d) and aging (Fig.2l-n). BuGZ-null MARCM clones and F/O lineage-tracing analyses of the RNAi clones also revealed that depression of BuGZ suppressed the ISC proliferation after clone induction (Fig.2h-k). Meanwhile, we indeed also overexpressed BuGZ-FL in the *esg*⁺ cells of young *Drosophila* and found an elevation of ISC proliferation rate (Fig.2l-n). BuGZ-FL-overexpressed MARCM clones were significantly bigger than the control ones (Fig. 2h,*

i). Based on these data, we concluded that BuGZ promotes ISC proliferation. In addition, we found that BuGZ depletion with two different RNAi lines in *esg*⁺ cells did not affect the number of *esg*⁺ (ISC and progenitor cells) or phospho-histone H3⁺ (pH3⁺; an immunomarker of cell proliferation) cells in young midguts without injury (at these conditions most ISCs are under quiescent state) (Fig. 2a-d). These data indicated that the depletion of BugZ did not cause ISCs reduction or increase. Moreover, the TUNEL assay proposed in BuGZ-null clones indicated that the depletion of BugZ did not cause ISC apoptosis (supplementary Fig.3m-n). By checking the cell-type-specific makers in the BuGZ-null clones, we found that the presence of *Dl*⁺ ISCs, *Pros*⁺ EEs, *NRE-LacZ*⁺ EBs, and polyploid ECs in BuGZ-null clones on day10 after clone induction (ACI), which proposed that BuGZ might be not required for ISC differentiation (Supplementary Fig. 3o-q). In the revised manuscript, we also stained Delta as the ISCs fate marker of midguts without or with BLM-induced injury. We found that depletion of BuGZ in midguts of young flies without injury (homeostasis conditions) did not affect the number of *Dl*⁺ ISCs (supplementary Fig.3c-d). This data suggest that BuGZ was not essential for the basic maintenance of ISCs. While, after injury (ISCs were activated to proliferation and differentiation), we indeed found that depletion of BuGZ led to a failure of elevation of *Dl*⁺ ISCs in the injured midguts (supplementary Fig.3c-d). This data suggested that BuGZ was essential for the activation of ISC proliferation to repair the damaged gut epithelia. Thus, we proposed that BuGZ was not required for ISC basic maintenance but was essential for ISC proliferation after injury.

*BuGZ depletion in *esg*+ cells affected the number of Dl+ cells in injured young midguts, but not control ones (supplementary Fig. 3c, d)*

2. The authors concluded that BuGZ mediates ISC proliferation and gut repair after injury by performing cell-autonomous functions. However, the authors did not provide clear evidence to support this claim. To strengthen their argument, they could demonstrate that RNAi knockdown of BuGZ in enteroendocrine cells or enteroblasts completely blocks BuGZ expression in ISCs. This would help to confirm that BuGZ functions in a cell-autonomous manner to regulate ISC proliferation and gut repair after injury.

Response:

*Thanks for this question. In our original manuscript, we showed that BuGZ RNAi in the ISC cells induced by ISC^{TS} (NRE-lacZ; *esg*-GAL4, UAS-GFP/CyO; *tub*-Gal80ts,*

NRE-Gal80) inhibited the proliferation of ISC after injury (Fig. 2d). However, *BuGZ* depletion in EBs induced by *NRE^{TS}* (*NRE-GAL4*, *UAS-GFP*, *tub-Gal80ts/CyO*), or in EEs by *386y-Gal4* had no effect on the proliferation of ISC upon gut damage (Fig. 2e-f). These data suggested that *BuGZ* mediated the proliferation of ISC and gut repair after injury by performing cell-autonomous functions. In our revised manuscript, to answer the question of this reviewer, we verified the *BuGZ* expression levels in ISCs (labeled with *Delta*) upon depleting *BuGZ* in ISCs, EBs or EEs. Consistent with our previous data, it showed *BuGZ* depletions in ISCs diminished the *BuGZ* level in *Delta*+ cells, whereas *BuGZ* RNAi in EEs or EBs did not affect *BuGZ* expression in *Delta*+ cells (supplementary Fig.3a, e-h). Thus, *BuGZ* mediates ISC proliferation and gut repair by performing cell-autonomous functions.

BuGZ protein level in *esg*+ cells upon depleting *BuGZ* in ISCs, EBs or EEs.

(supplementary Fig. 3e-h)

3. In Supplementary Fig. 3j, is the *NRE-LacZ* expressed in polyploid cells in the control?

Response:

Many thanks for this question. NRE-LacZ was typically expressed in all EBs and some polyploidy pre-ECs. Thus, some polyploidy pre-ECs were also NRE-LacZ+ cells in our data (supplementary Fig.3q).

4. The authors state that "increased expression of BuGZ led to ISC hyperproliferation." However, the data presented in Figure 2N show that the number of cells positive for the proliferation marker PH3 is less than 35 in BuGZ-overexpressing ISCs. Therefore, the term "hyperproliferation" may not be appropriate. Instead, the authors could state that BuGZ overexpression leads to an increase in ISC proliferation.

Response:

Thanks for this suggestion. In the revised manuscript, we changed the term "hyperproliferation" into "proliferation".

5. The relationship between BuGZ and YT521-B is not clear in the manuscript. To strengthen the authors' claim that BuGZ regulates YT521-B expression, the authors could perform a chromatin immunoprecipitation (ChIP) experiment to examine whether BuGZ can bind to the YT521-B promoter region. This would provide direct evidence of the interaction between BuGZ and YT521-B.

Response:

We appreciate this great suggestion for improving the integrity of our manuscript. According to this advice, in the revised manuscript, we have performed the assay of Chip-qPCR to examine whether BuGZ interacted with the YT521-B promoter region (P⁸ fragment). The results indicated that BuGZ bound to the YT521-B promoter region in both young and old flies (Fig. 5a).

Results of Chip-qPCR with immunoprecipitating BuGZ in young and old flies (Fig. 5a)

6. The mechanism by which BuGZ and YT521-B activate the MAPK pathway is critical for explaining the function of BuGZ and YT521-B in regulating ISC proliferation and gut repair after injury. The authors should provide more details on how BuGZ and YT521-B activate the MAPK pathway. This information would help to elucidate the downstream signaling pathways that are regulated by BuGZ and YT521-B and provide a more complete understanding of their roles in ISC proliferation and gut repair after injury.

Response:

We are grateful for this constructive suggestion. To study the detailed mechanism of the YT521-B modulation in the MAPK pathway, we examined the ERK activity assessed by immunostaining against the double phosphorylated ERK kinase (dpERK), which behaved as the key downstream target of the MAPK pathway⁶. Our data showed YT521-B overexpression inhibited the ERK activation caused by BuGZ overexpression (supplementary Fig. 7h). These data inferred that BuGZ-YT521-B axis regulated the MAPK pathway by modulating the ERK activity.

YT521-B regulates the MAPK pathway by ERK signal (supplementary Fig. 7h)

Minor points:

1. “BuGZ^{13S} overexpression in *esg*⁺ cells showed fewer effects on the ISC proliferation than BuGZ^{FL} (Fig. 3b, f, g)” in P8. It should be rephrased.

Response:

*Thanks for this suggestion. In the revised manuscript, we rephrased this sentence as “BuGZ^{13S} overexpression in *esg*⁺ cells showed fewer number of *esg*⁺ and pH3⁺ cells, compared with BuGZ^{FL} overexpression (Fig. 3b, f, g)”.*

2. For “contrary to the BuGZ RNAi phenotype that we demonstrated in the asymmetric cell division of ISC after injury” in P6, the sentence is confusing.

Response:

*We are sorry for not stating it clearly. In the revised manuscript, we have corrected it as “Previous studies have shown that BuGZ depletion blocked prometaphase-metaphase transition and elevated the mitotic index in the symmetric division of tumor cells. Conversely, we found that BuGZ RNAi reduced mitotic index (indicated by decreased *esg*⁺ cells and pH3⁺ cells in the midguts) in the asymmetric cell division of ISC after injury.”*

Reply to Reviewer #3

The manuscript entitled “Phase separation regulates gut regeneration and aging by m6A regulators” authored by Zhang Q. et al. describes the molecular mechanism of how liquid-liquid phase separation plays a role in the regeneration and aging of gut in fly. They focused on BuGZ protein, which had been known to form a liquid-like condensate in mitotic cells, and examined its role in intestinal stem cells. They performed a number of experiments and found that BuGZ forms a condensate in interphase intestinal stem cell nucleus and affects fly gut repair and longevity. They also identified down-stream factors such as YT512-B and demonstrated that BuGZ play a role in the transcriptional regulation of YT521-B and Ime4/Mettl14 genes. From mechanistic point of view, the manuscript presents a new pathway and mechanism of BuGZ-dependent gut regeneration. On the other hand, in terms of a role of liquid-liquid phase separation in this process, most of the data are still descriptive and do not provide new mechanism or concept of phase separation in gut repair. If the authors are willing to show a critical role of BuGZ “phase behavior” in gut repair (this is what they emphasize in Introduction), they have to clarify the following issues.

Response:

Many thanks for this reviewer's questions. We believe the revisions we made as detailed below have addressed all the concerns raised.

1. Because BuGZ had been identified as a microtubule-regulating protein, it was surprising that BuGZ was involved in the transcriptional regulation. As shown in Fig. 1a and 1j (and many others), BuGZ formed several huge condensates in ISC nuclei. It seems that the size, shape and number of foci in a nucleus varied a lot from cells to cells. If BuGZ is involved in the transcriptional regulation, how a large variation of the size and number of foci can be explained? It is more reasonable to expect a number of target genes not only YT521-B should exist. If the authors wants to understand the role of BuGZ condensate in the transcriptional regulation, they have to clarify this. I wonder

if they ever performed chromatin immunoprecipitation to identify all possible target genes.

Response:

We appreciate this great suggestion for improving the integrity of our manuscript. According to previous study, BuGZ was identified as a transcription factor to control self-renewal and pluripotency of hESCs⁷. In this study, gene ontology analysis of the genes that were directly regulated by BuGZ (overlaps of ChIP-Seq bound genes with differentially expressed genes identified from RNA-Seq), indicated that BuGZ mediated the cell proliferation and division in hESCs⁷. Based on these data, we inferred there might be a bunch of target genes beyond YT521-B under the regulation of BuGZ condensate in ISCs, which might result in the large variation of the size and number of BuGZ foci in the nuclei of ISCs. Moreover, focused on the genes involved in m⁶A modification, we noticed that BuGZ bound to the DNA fragments of several m⁶A regulators in their Chip-Seq data, including Mettl16 (“m⁶A writer”), ALKBH5 (“m⁶A eraser”), and “m⁶A readers” (HNRNPA1, HNRNPA3, IGF2BP1) (figure A-E). It implied that BuGZ might interact with the DNA regions of other m⁶A regulators beyond YT521-B in ISCs. To confirm this, we performed Chip-qPCR to verify whether BuGZ bound to the promoter regions of m⁶A-related genes in Drosophila including Ime4, Mettl14, Fl(2)D, Nito, Vir, YT521-B, CG6422. The results showed that BuGZ did not interact with Ime4 or Mettl14, whereas bound to the promoters of YT521-B (Fig.5a) and vir, CG6422 (figure F). Altogether, we conclude that BuGZ condensate regulated various targets genes in ISCs of Drosophila.

BuGZ bound to the DNA fragments of several m^6A regulators including Mettl16, ALKBH5, HNRNPA1, HNRNPA3, IGF2BP1 (figure A-E).

Chip-qPCR was carried out to verify whether BuGZ bound to the promoter regions of m^6A -related genes in Drosophila including Ime4, Mettl14, Fl(2)D, Nito, Vir, CG6422 (figure F).

Results of Chip-qPCR with immunoprecipitating BuGZ in young and old flies (Fig. 5a).

2. The authors proposed that BuGZ binds to the promoter region of YT521-B gene, assemble there, and finally form a large condensate. Then, I wonder what is the physiological significance of the formation of such a large condensate, rather than a simple “binding”. Generally, transcriptional control can be achieved by simple binding of a single or a few copies of a transcription factor to the promoter. If the authors want to understand the role of BuGZ phase separation, they have to show the evidence for physiological role of condensate formation, which cannot be achieved by a simple binding.

Response:

We appreciate for these great comments. To further study the role of YT521-B promoter region on BuGZ phase separation beyond the simple physical binding, we performed the FRAP assay with or without adding the P⁸ fragment into the purified scarlet-BuGZ. The results revealed that the promoter region (P⁸) of YT521-B inhibited the fluorescence recovery of BuGZ droplets (supplementary Fig. 8i-l). Thus, the phenomenon that BuGZ’s interaction with the YT521-B promoter reduced the dynamicity of BuGZ condensate, implied that YT521-B promoter might recruit the BuGZ condensate to stably bind to the transcription region to constantly exert its transcriptional function.

FARP of scarlet-BuGZ with or without adding P⁸ fragment (supplementary Fig. 8i-l).

3. The morphological variation of BuGZ condensates in a cell reminds various nuclear speckles rather than transcription condensates, which are generally much smaller. The authors examined only PML bodies with anti-SUMO antibody. However, I wonder whether BuGZ condensate colocalize with other known nuclear speckles. Because BuGZ has been demonstrated to be involved in mRNA splicing (as they mentioned in Introduction), this may be possible. Although I understand that the BuGZ condensate is huge and not similar to any known nuclear speckles, these possibilities should be tested.

Response:

We appreciate this constructive comment and suggestion. To further explored the nuclear localization of BuGZ condensate in ISCs, we speculated whether BuGZ puncta co-localized with other nuclear membraneless organelles showing the LLPS behavior in previous reports⁸⁻¹¹ including chromatin, heterochromatin and DNA repair foci. Consequently, we found that BuGZ puncta was partially co-located with the TriMethyl-Histone H3-K27-labeled chromatin, but not the HPI α -labeled heterochromatin, or γ H₂AvD-labeled DNA repair foci. We have added these new data in the revised manuscript (Fig. 1p-r). Beyond these five nuclear membraneless structures, we additionally observed whether BuGZ foci localized with another two nuclear

membraneless organelles including SC35-labeled nuclear speckle and cajal body with probe that exhibited the property of phase separation as well^{8,9}. However, the extensive homology SC35 antibodies (abcam: ab11826; sigma: S4045; santa cruz: sc-53518) did not work in *Drosophila* guts, and the cajal body probes designed either by the published paper¹² or the probe company could not specifically identify the *Drosophila* cajal body in guts.

Localizations of BuGZ foci with other nuclear granules in ISCs (Fig. 1n-r).

References

1. Jiang, H. *et al.* A microtubule-associated zinc finger protein, BuGZ, regulates mitotic chromosome alignment by ensuring Bub3 stability and kinetochore targeting. *Dev Cell* **28**, 268-281 (2014).
2. Wan, Y. *et al.* Splicing function of mitotic regulators links R-loop-mediated DNA damage to tumor cell killing. *J Cell Biol* **209**, 235-246 (2015).
3. Guo, Z. & Ohlstein, B. Stem cell regulation. Bidirectional Notch signaling regulates *Drosophila* intestinal stem cell multipotency. *Science* **350** (2015).

4. Wang, L., Zeng, X., Ryoo, H.D. & Jasper, H. Integration of UPRER and oxidative stress signaling in the control of intestinal stem cell proliferation. *PLoS Genet* **10**, e1004568 (2014).
5. McGuire, S.E., Roman, G. & Davis, R.L. Gene expression systems in *Drosophila*: a synthesis of time and space. *Trends Genet* **20**, 384-391 (2004).
6. Biteau, B. & Jasper, H. EGF signaling regulates the proliferation of intestinal stem cells in *Drosophila*. *Development* **138**, 1045-1055 (2011).
7. Fang, F. *et al.* A distinct isoform of ZNF207 controls self-renewal and pluripotency of human embryonic stem cells. *Nat Commun* **9**, 4384 (2018).
8. Boija, A., Klein, I.A. & Young, R.A. Biomolecular Condensates and Cancer. *Cancer Cell* **39**, 174-192 (2021).
9. Grosch, M., Ittermann, S., Shaposhnikov, D. & Drukker, M. Chromatin-Associated Membraneless Organelles in Regulation of Cellular Differentiation. *Stem Cell Reports* **15**, 1220-1232 (2020).
10. Sanulli, S. *et al.* HP1 reshapes nucleosome core to promote phase separation of heterochromatin. *Nature* **575**, 390-394 (2019).
11. Gibson, B.A. *et al.* Organization of Chromatin by Intrinsic and Regulated Phase Separation. *Cell* **179**, 470-484 e421 (2019).
12. Liu, J.L. *et al.* The *Drosophila melanogaster* Cajal body. *J Cell Biol* **172**, 875-884 (2006).

REVIEWER COMMENTS

Reviewer #1 (Remarks to the Author):

The revision has addressed most of the questions in my review. Some minor issues:

In the field, m6A is referred to N6-methyladenosine, not N6-methyladenine (page 3).

Another thing is that the work by Du et al, Dev. Cell, 2022 Aug 22;57(16):1976-1994.e8. doi: 10.1016/j.devcel.2022.07.006 is highly related to this manuscript, and should be included in the Discussion, as it directly assesses Mettl14 functions in intestinal stem cells.

Reviewer #2 (Remarks to the Author):

Reviewers 1 and I have raised concerns regarding the regulation of the MAPK pathway by BuGZ and YT521-B. However, the evidence provided in supplementary Fig. 7h is not convincing enough to support the claim that BuGZ overexpression leads to an increase in dpErk. Additionally, the inhibition of this increase by YT521-B overexpression is also not adequately supported. As a result, it is critical to consider alternative explanations and conduct additional investigations to gain further insights into the role of BuGZ and YT521-B in the MAPK pathway.

Reviewer #3 (Remarks to the Author):

The manuscript entitled “Phase separation regulates gut regeneration and aging by m6A regulators” authored by Chang Q. et al, has been greatly improved after the revision. However, I'm afraid some of the points that I raised have not been fully clarified yet. Here are the concerns which remain unclarified.

Comment #1. Because BuGZ had been identified as a microtubule-regulating protein, it was surprising that BuGZ was involved in the transcriptional regulation. As shown in Fig. 1a and 1j (and many others), BuGZ formed several huge condensates in ISC nuclei. It seems that the size, shape and number of foci in a nucleus varied a lot from cells to cells. If BuGZ is involved in the transcriptional regulation, how a large variation of the size and number of foci can be explained? It is more reasonable to expect a number of target genes not only YT521-B should exist. If the authors want to understand the role of BuGZ

condensate in the transcriptional regulation, they have to clarify this. I wonder if they ever performed chromatin immunoprecipitation to identify all possible target genes.

Response:

We appreciate this great suggestion for improving the integrity of our manuscript. According to previous study, BuGZ was identified as a transcription factor to control self-renewal and pluripotency of hESCs⁷. In this study, gene ontology analysis of the genes that were directly regulated by BuGZ (overlaps of ChIP-Seq bound genes with differentially expressed genes identified from RNA-Seq), indicated that BuGZ mediated the cell proliferation and division in hESCs⁷. Based on these data, we inferred there might be a bunch of target genes beyond YT521-B under the regulation of BuGZ condensate in ISCs, which might result in the large variation of the size and number of BuGZ foci in the nuclei of ISCs. Moreover, focused on the genes involved in m6A modification, we noticed that BuGZ bound to the DNA fragments of several m6A regulators in their Chip-Seq data, including Mettl16 (“m6A writer”), ALKBH5 (“m6A eraser”), and “m6A readers” (HNRNPA1, HNRNPA3, IGF2BP1) (figure A-E). It implied that BuGZ might interact with the DNA regions of other m6A regulators beyond YT521-B in ISCs. To confirm this, we performed Chip-qPCR to verify whether BuGZ bound to the promoter regions of m6A-related genes in Drosophila including Ime4, Mettl14, Fl(2)D, Nito, Vir, YT521-B, CG6422. The results showed that BuGZ did not interact with Ime4 or Mettl14, whereas bound to the promoters of YT521-B (Fig.5a) and vir, CG6422 (figure F). Altogether, we conclude that BuGZ condensate regulated various targets genes in ISCs of Drosophila.

> The authors integrated previously published information on the genes that are regulated by BuGZ and conducted Chip-qPCR to confirm some of them. This information is very important and highly valuable in understanding the mechanism of BuGZ function. On the other hand, the authors did not give clear answer to my question “If BuGZ is involved in the transcriptional regulation, how a large variation of the size and number of foci can be explained?” This comment is more or less related to my next comment (comment #2). If BuGZ regulates a number of genes, what is the physiological significance of forming various sizes and numbers of condensates in a nucleus; in some cells, they form a single big chunk of condensate, but in other cells, they form several small condensates. As this paper is focusing on the formation of BuGZ condensate, the authors have to demonstrate physiological significance of the condensate formation over simple stoichiometric binding.

Comment #2. The authors proposed that BuGZ binds to the promoter region of YT521-B gene, assemble there, and finally form a large condensate. Then, I wonder what is the physiological significance of the formation of such a large condensate, rather than a simple “binding”. Generally, transcriptional control can be achieved by simple binding of a single or a few copies of a transcription factor to the promoter. If

the authors want to understand the role of BuGZ phase separation, they have to show the evidence for physiological role of condensate formation, which cannot be achieved by a simple binding.

Response:

We appreciate for these great comments. To further study the role of YT521-B promoter region on BuGZ phase separation beyond the simple physical binding, we performed the FRAP assay with or without adding the P8 fragment into the purified scarlet-BuGZ. The results revealed that the promoter region (P8) of YT521-B inhibited the fluorescence recovery of BuGZ droplets (supplementary Fig. 8i-l). Thus, the phenomenon that BuGZ's interaction with the YT521-B promoter reduced the dynamicity of BuGZ condensate, implied that YT521-B promoter might recruit the BuGZ condensate to stably bind to the transcription region to constantly exert its transcriptional function.

> The authors performed FRAP analysis to show that the BuGZ condensate in the nucleus is "liquid-like state". This is important result and should be appreciated. However, they did not give clear answer to my concern 'what is the physiological significance of the formation of such a large condensate, rather than a simple "binding"'. This is closely related to my comment #1. Given that BuGZ regulates a number of genes, why does it form a huge condensate in a nucleus rather than bind to individual promoters stoichiometrically? This manuscript will be greatly improved if they reveal it. I still have a strong impression that the manuscript presents a new pathway and mechanism of BuGZ-dependent gut regeneration from mechanistic point of view. However, in terms of a role of LLPS, most of the data are still descriptive and do not provide new mechanism or concept of phase separation. If the authors are willing to show a critical role of BuGZ "phase behavior" in gut repair (this is what they emphasize in Introduction), they have to clarify this issue.

Reply to Reviewer #1

The revision has addressed most of the questions in my review. Some minor issues: In the field, m6A is referred to N6-methyladenosine, not N6-methyladenine (page 3). Another thing is that the work by Du et al, Dev. Cell, 2022 Aug 22;57(16):1976-1994.e8. doi: 10.1016/j.devcel.2022.07.006 is highly related to this manuscript, and should be included in the Discussion, as it directly assesses Mettl14 functions in intestinal stem cells.

Response:

Thank you again for your review work and valuable suggestions. We feel quite sorry about the typo in the former version. We have corrected the N6-methyladenine with N6-methyladenosine in our revised manuscript. Furthermore, we have emphasized the function of Mettl14 in intestinal stem cells and added the sentence “Mettl14-mediated GsdmC N6-adenomethylation is essential for the survival of Lgr5⁺ intestinal stem cells and the maintenance of normal colonic epithelial regeneration^{49,1}” into the “Discussion” section of the revised manuscript.

Reply to Reviewer #2

Reviewers 1 and I have raised concerns regarding the regulation of the MAPK pathway by BuGZ and YT521-B. However, the evidence provided in Supplementary Fig. 7h is not convincing enough to support the claim that BuGZ overexpression leads to an increase in dpErk. Additionally, the inhibition of this increase by YT521-B overexpression is also not adequately supported. As a result, it is critical to consider

alternative explanations and conduct additional investigations to gain further insights into the role of BuGZ and YT521-B in the MAPK pathway.

Response:

*We really appreciate your valuable suggestion to improve our understanding of the role of BuGZ and YT521-B on the MAPK pathway. To Address this issue, we conducted additional assays to confirm that the MAPK signaling pathway was regulated by BuGZ-YT521-B axis to modulate ISC proliferation. Firstly, we observed that BuGZ depletion in ISCs resulted in a significant decrease of dpERK signal in ISCs (Supplementary Fig. 7i). Moreover, the induction of YT521-B RNAi restored the down-regulation of dpERK in ISCs caused by BuGZ depletion (Supplementary Fig. 7i). To provide more evidence that the BuGZ-YT521-B axis regulates MAPK signaling during ISC proliferation, we also investigated the impact of BuGZ and YT521-B on the mRNA expressions of Argos (aos), Kekk1 (kek1), and Sprouty (sty) in sorted *esg*⁺ cells, all of which were reported to act as downstream targets of MAPK signaling pathway²⁻⁵. We found that the overexpression of BuGZ^{FL} augmented the mRNA levels of *kek1*, *sty*, and *aos*, indicating the elevation of the MAPK signaling activity²⁻⁵ (Supplementary Fig. 7j). Accordingly, the overexpression of YT521-B reversed the increased mRNA levels of *kek1*, *sty*, and *aos* induced by BuGZ^{FL} overexpression (Supplementary Fig. 7j). Moreover, YT521-B RNAi rescued the reduction of *aos*, *kek1*, and *sty* mRNA levels caused by BuGZ RNAi during gut regeneration (Supplementary Fig. 7k). Thus, all these findings revealed that the BuGZ-YT521-B axis played an important role in regulating the MAPK signaling activity during ISC proliferation.*

Supplementary Fig. 7i

Supplementary Fig. 7j

Supplementary Fig. 7k

Reply to Reviewer #3

Comment #1

The manuscript entitled “Phase separation regulates gut regeneration and aging by m6A regulators” authored by Chang Q. et al, has been greatly improved after the revision. However, I'm afraid some of the points that I raised have not been fully clarified yet. Here are the concerns which remain unclarified.

Comment #1. Because BuGZ had been identified as a microtubule-regulating protein, it was surprising that BuGZ was involved in the transcriptional regulation. As shown in Fig. 1a and 1j (and many others), BuGZ formed several huge condensates in ISC nuclei. It seems that the size, shape and number of foci in a nucleus varied a lot from cells to cells. If BuGZ is involved in the transcriptional regulation, how a large variation of the

size and number of foci can be explained? It is more reasonable to expect a number of target genes not only YT521-B should exist. If the authors want to understand the role of BuGZ condensate in the transcriptional regulation, they have to clarify this. I wonder if they ever performed chromatin immunoprecipitation to identify all possible target genes.

Response:

We appreciate this great suggestion for improving the integrity of our manuscript. According to previous study, BuGZ was identified as a transcription factor to control self-renewal and pluripotency of hESCs⁷. In this study, gene ontology analysis of the genes that were directly regulated by BuGZ (overlaps of ChIP-Seq bound genes with differentially expressed genes identified from RNA-Seq), indicated that BuGZ mediated the cell proliferation and division in hESCs⁷. Based on these data, we inferred there might be a bunch of target genes beyond YT521-B under the regulation of BuGZ condensate in ISCs, which might result in the large variation of the size and number of BuGZ foci in the nuclei of ISCs. Moreover, focused on the genes involved in m⁶A modification, we noticed that BuGZ bound to the DNA fragments of several m⁶A regulators in their Chip-Seq data, including Mettl16 (“m⁶A writer”), ALKBH5 (“m⁶A eraser”), and “m⁶A readers” (HNRNPA1, HNRNPA3, IGF2BP1) (figure A-E). It implied that BuGZ might interact with the DNA regions of other m⁶A regulators beyond YT521-B in ISCs. To confirm this, we performed Chip-qPCR to verify whether BuGZ bound to the promoter regions of m⁶A-related genes in *Drosophila* including

Ime4, Mettl14, Fl(2)D, Nito, Vir, YT521-B, CG6422. The results showed that BuGZ did not interact with Ime4 or Mettl14, whereas bound to the promoters of YT521-B (Fig.5a) and vir, CG6422 (figure F). Altogether, we conclude that BuGZ condensate regulated various targets genes in ISCs of Drosophila.

> The authors integrated previously published information on the genes that are regulated by BuGZ and conducted Chip-qPCR to confirm some of them. This information is very important and highly valuable in understanding the mechanism of BuGZ function. On the other hand, the authors did not give clear answer to my question “If BuGZ is involved in the transcriptional regulation, how a large variation of the size and number of foci can be explained?” This comment is more or less related to my next comment (comment #2). If BuGZ regulates a number of genes, what is the physiological significance of forming various sizes and numbers of condensates in a nucleus; in some cells, they form a single big chunk of condensate, but in other cells, they form several small condensates. As this paper is focusing on the formation of BuGZ condensate, the authors have to demonstrate physiological significance of the condensate formation over simple stoichiometric binding.

Response:

Thank you for your positive feedback and valuable suggestions to improve the quality of our manuscript. Here, we provided more possible clarifications to explain the variation of the size and number of BuGZ foci, and performed extra assays to comprehensively demonstrate the physiological significance of the BuGZ condensation.

First, BuGZ expression level contributed to the variation of the condensate size and number. For instance, endogenous BuGZ generally formed multiple, relatively uniform and small foci in aged and injured ISCs, as depicted in Fig. 1a, i and Supplementary Fig.3e, whereas BuGZ overexpression led to the emergence of big chunks of condensates beyond the small ones in ISCs, as shown in Fig. 1j, Fig. 3b,3m-n, Fig.5f and Fig 6a,6e. Despite BuGZ overexpressed cells seemed to simply exhibit one or several big chunks of condensates, there were also numerous and small BuGZ puncta within the nuclei that were not easily visible due to their weaker fluorescence intensity when compared with the big ones. Second, the variation of condensate number and size in the nucleus might be also attributed to the diverse target genes regulated by the BuGZ condensate, which was supported by the previously published information⁶ and our Chip-qPCR analyses in our previous response manuscript (figure F).

To further understand the physiological significance of the BuGZ condensate formation, we investigated whether BuGZ condensation modulated YT521-B transcription. As BuGZ-13S protein whose 6 tyrosines (Y) and 7 phenylalanines (F) were replaced by serines disrupted the phase separation property of BuGZ as previously reported⁷, our results demonstrated that the overexpression of BuGZ-FL, in contrast to BuGZ-13S, inhibited the mRNA level of YT521-B (Supplementary Fig. 6e). Besides, since BuGZ peptide C (BuGZ-C) in the IDR region, different from CS (BuGZ-CS) dramatically suppressed the coacervation of wild-type BuGZ in vitro⁷ and in vivo (Fig. 3m,3n), BuGZ-C overexpression increased the YT521-B mRNA level upon gut injury, whereas BuGZ-CS overexpression showed compromised effect on prompting YT521-B

expression level when compared with BuGZ-C (Supplementary Fig. 6f). Additionally, our luciferase assays manifested that the transcriptional activity of YT521-B was suppressed by BuGZ-FL overexpression, when compared with overexpressing BuGZ-13S (Supplementary Fig. 6h). Furthermore, the Chip-qPCR data indicated that BuGZ-FL exhibited enhanced binding to the YT521-B promoter in contrast to BuGZ-13S (Supplementary Fig. 6i). Meanwhile, BuGZ-C overexpression suppressed the binding of the YT521-B promoter to BuGZ in Flag-BuGZ knock-in flies under BLM treatment, whereas BuGZ-CS overexpression displayed diminished effect on attenuating the binding of BuGZ to the YT521-B promoter in relative to BuGZ-C (Supplementary Fig. 6j). Thus, these findings revealed the physiological significance of the BuGZ condensate formation in DNA binding and transcriptional regulation during ISC proliferation and gut repair, and provided insights into the regulatory mechanisms underlying the BuGZ-mediated modulation of gene expression. We have added this part into the “Discussion” section of the revised manuscript.

Supplementary Fig. 6e-f

Supplementary Fig. 6h

Supplementary Fig. 6i-j

Comment #2

The authors proposed that BuGZ binds to the promoter region of YT521-B gene, assemble there, and finally form a large condensate. Then, I wonder what is the physiological significance of the formation of such a large condensate, rather than a

simple “binding”. Generally, transcriptional control can be achieved by simple binding of a single or a few copies of a transcription factor to the promoter. If the authors want to understand the role of BuGZ phase separation, they have to show the evidence for physiological role of condensate formation, which cannot be achieved by a simple binding.

Response:

We appreciate for these great comments. To further study the role of YT521-B promoter region on BuGZ phase separation beyond the simple physical binding, we performed the FRAP assay with or without adding the P8 fragment into the purified scarlet-BuGZ. The results revealed that the promotor region (P8) of YT521-B inhibited the fluorescence recovery of BuGZ droplets (Supplementary Fig. 8i-l). Thus, the phenomenon that BuGZ’s interaction with the YT521-B promotor reduced the dynamicity of BuGZ condensate, implied that YT521-B promotor might recruit the BuGZ condensate to stably bind to the transcription region to constantly exert its transcriptional function.

> The authors performed FRAP analysis to show that the BuGZ condensate in the nucleus is “liquid-like state”. This is important result and should be appreciated. However, they did not give clear answer to my concern ‘what is the physiological significance of the formation of such a large condensate, rather than a simple “binding”’. This is closely related to my comment #1. Given that BuGZ regulates a number of genes,

why does it form a huge condensate in a nucleus rather than bind to individual promoters stoichiometrically? This manuscript will be greatly improved if they reveal it. I still have a strong impression that the manuscript presents a new pathway and mechanism of BuGZ-dependent gut regeneration from mechanistic point of view. However, in terms of a role of LLPS, most of the data are still descriptive and do not provide new mechanism or concept of phase separation. If the authors are willing to show a critical role of BuGZ “phase behavior” in gut repair (this is what they emphasize in Introduction), they have to clarify this issue.

Response:

We really appreciate this great suggestion for improving the integrity of our manuscript. We hope that the issues about the size and physiological significance of BuGZ condensates have been properly addressed in the response of Comment 1.

In our previous revised manuscript, we observed that BuGZ formed nuclear condensates in PQ/BLM-treated ISCs along with its elevated mRNA and protein levels (Fig. 1e-i). To explore whether LLPS of BuGZ contributed to ISC proliferation during gut repair and regeneration, we overexpressed BuGZ-C/CS and found that BuGZ-C overexpression was more effective than BuGZ-CS in inhibiting ISC proliferation under BLM treatment conditions (Fig. 3j-l).

In our current revised manuscript, to further show the critical role of BuGZ condensate in gut repair, we analyzed the behavior of BuGZ puncta in injured ISCs upon overexpressing the truncated BuGZ-C or BuGZ-CS. Compared to the control group, overexpression of BuGZ-C inhibited the phase separation of BuGZ during gut

repair (Fig. 3m). BuGZ-C exhibited more effective influence than BuGZ-CS on suppressing phase separation of BuGZ (Fig.3m). Furthermore, since integrity of gut epithelial barrier and length of midgut were associated with epithelial turnover^{8, 9}, we assessed the integrity and length of the intestine in BuGZ-C/CS overexpressed ISCs. Armadillo immunostaining assay, which was used to test integrity of gut epithelium⁹, indicated that overexpression of BuGZ-C induced unrecoverable injury to the intestinal epithelium (Supplementary Fig.4k), which also resulted in shortened length of the midguts (Supplementary Fig.4l-m). Additionally, overexpression of BuGZ-C caused higher mortality following BLM-induced chronic injury (Supplementary Fig.4o). Besides, BuGZ-CS showed compromised effects on the intestinal integrity, length and lifespan when compared with BuGZ-C (Supplementary Fig.4k-m and Supplementary Fig.4o). Based on these findings, we concluded that BuGZ phase separation played a crucial role on promoting intestinal integrity during gut repair.

Fig. 3m

Supplementary Fig.4k

Supplementary Fig.4l-m

Supplementary Fig.4o

Reference

1. Du, J. *et al.* N(6)-adenomethylation of GsdmC is essential for Lgr5(+) stem cell survival to maintain normal colonic epithelial morphogenesis. *Dev Cell* **57**, 1976-1994 e1978 (2022).
2. Golembo M, Schweitzer R, Freeman M & BZ., S. argos transcription is induced by the Drosophila EGF receptor pathway to form an inhibitory feedback loop. *Development* **122**, 223-230 (1996).
3. Kramer S, O.M., Hacohen N, Krasnow MA, Hiromi Y Sprouty a common antagonist of FGF and EGF signaling pathways in Drosophila. *Development* **126**, 2515-2525 (1999).
4. Shilo, B.Z. The regulation and functions of MAPK pathways in Drosophila. *Methods* **68**, 151-159 (2014).
5. Ghiglione C, C.K.r., Amundadottir LT, Boswell RE, Perrimon N, Duffy JB. The Transmembrane Molecule Kekkon 1 Acts in a Feedback Loop to Negatively Regulate

- the Activity of the Drosophila EGF Receptor during Oogenesis. *Cell* **96**, 847-856 (1999).
6. Fang, F. *et al.* A distinct isoform of ZNF207 controls self-renewal and pluripotency of human embryonic stem cells. *Nat Commun* **9**, 4384 (2018).
 7. Jiang, H. *et al.* Phase transition of spindle-associated protein regulate spindle apparatus assembly. *Cell* **163**, 108-122 (2015).
 8. Mundorf, J., Donohoe, C.D., McClure, C.D., Southall, T.D. & Uhlirova, M. Ets21c Governs Tissue Renewal, Stress Tolerance, and Aging in the Drosophila Intestine. *Cell Rep* **27**, 3019-3033 e3015 (2019).
 9. Lee, S.H., Goo, T.-W. & Yun, E.-Y. Allomyrina dichotoma larval extract has protective effects against gut permeability of dextran sulfate sodium-fed Drosophila by E-cadherin and armadillo. *Journal of Ethnopharmacology* **279** (2021).

REVIEWERS' COMMENTS

Reviewer #2 (Remarks to the Author):

No more comments

Reviewer #3 (Remarks to the Author):

It seems that the concerns that I raised for the revised manuscript have been clarified. I think the manuscript is now more convincing.